# DARA: Dynamics-Aware Reward Augmentation in Offline Reinforcement Learning

**Jinxin Liu**[123*]  **Hongyin Zhang**[1*]  **Donglin Wang**[13†]
[1] Westlake University.   [2] Zhejiang University.
[3] Institute of Advanced Technology, Westlake Institute for Advanced Study.
{liujinxin, zhanghongyin, wangdonglin}@westlake.edu.cn

## Abstract

Offline reinforcement learning algorithms promise to be applicable in settings where a fixed dataset is available and no new experience can be acquired. However, such formulation is inevitably offline-data-hungry and, in practice, collecting a large offline dataset for one specific task over one specific environment is also costly and laborious. In this paper, we thus 1) formulate the offline dynamics adaptation by using (source) offline data collected from another dynamics to relax the requirement for the extensive (target) offline data, 2) characterize the dynamics shift problem in which prior offline methods do not scale well, and 3) derive a simple dynamics-aware reward augmentation (DARA) framework from both model-free and model-based offline settings. Specifically, DARA emphasizes learning from those source transition pairs that are adaptive for the target environment and mitigates the offline dynamics shift by characterizing state-action-next-state pairs instead of the typical state-action distribution sketched by prior offline RL methods. The experimental evaluation demonstrates that DARA, by augmenting rewards in the source offline dataset, can acquire an adaptive policy for the target environment and yet significantly reduce the requirement of target offline data. With only modest amounts of target offline data, our performance consistently outperforms the prior offline RL methods in both simulated and real-world tasks.

## 1 Introduction

Offline reinforcement learning (RL) (Levine et al., 2020; Lange et al., 2012), the task of learning from the previously collected dataset, holds the promise of acquiring policies without any costly active interaction required in the standard online RL paradigm. However, we note that although the active trail-and-error (online exploration) is eliminated, the performance of offline RL method heavily relies on the amount of offline data that is used for training. As shown in Figure 1, the performance deteriorates dramatically as the amount of offline data decreases. A natural question therefore arises: can we reduce the amount of the (target) offline data without significantly affecting the final performance for the target task?

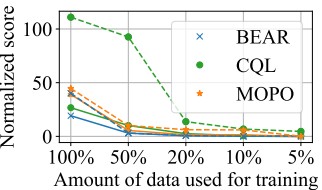

Figure 1: Solid and dashed lines denote offline Medium-Replay and Medium-Expert data in D4RL (Walker2d) *resp*.

Bringing the idea from the transfer learning (Pan & Yang, 2010), we assume that we have access to another (source) offline dataset, hoping that we can leverage this dataset to compensate for the performance degradation caused by the reduced (target) offline dataset. In the offline setting, previous work (Siegel et al., 2020; Chebotar et al., 2021) has characterized the reward (goal) difference between the source and target, relying on the "conflicting" or multi-goal offline dataset (Fu et al., 2020), while we focus on the relatively unexplored transition dynamics difference between the source dataset and the target environment. Meanwhile, we believe that this dynamics shift is not arbitrary in reality: in healthcare treatment, offline data for a particular patient is often limited, whereas we can obtain diagnostic data from other patients with the same case (same reward/goal)

---

*Equal contribution.
†Corresponding author.

and there often exist individual differences between patients (source dataset with different transition dynamics). Careful treatment with respect to the individual differences is thus a crucial requirement.

Given source offline data, the main challenge is to cope with the transition dynamics difference, *i.e.*, strictly tracking the state-action supported by the source offline data can not guarantee that the same transition (state-action-next-state) can be achieved in the target environment. However, in the offline setting, such dynamics shift is not explicitly characterized by the previous offline RL methods, where they typically attribute the difficulty of learning from offline data to the state-action distribution shift (Chen & Jiang, 2019; Liu et al., 2018). The corresponding algorithms (Fujimoto et al., 2019; Abdolmaleki et al., 2018; Yu et al., 2020) that model the support of state-action distribution induced by the learned policy, will inevitably suffer from the transfer problem where dynamics shift happens.

Our approach is motivated by the well established connection between reward modification and dynamics adaptation (Kumar et al., 2020b; Eysenbach & Levine, 2019; Eysenbach et al., 2021), which indicates that, by modifying rewards, one can train a policy in one environment and make the learned policy to be suitable for another environment (with different dynamics). Thus, we propose to exploit the *joint distribution of state-action-next-state*: besides characterizing the state-action distribution shift as in prior offline RL algorithms, we additionally identify the dynamics (*i.e.*, the conditional distribution of next-state given current state-action pair) shift and penalize the agent with a dynamics-aware reward modification. Intuitively, this reward modification aims to discourage the learning from these offline transitions that are likely in source but are unlikely in the target environment. Unlike the concurrent work (Ball et al., 2021; Mitchell et al., 2021) paying attention to the offline domain generalization, we explicitly focus on the offline domain (dynamics) adaptation.

Our principal contribution in this work is the characterization of the dynamics shift in offline RL and the derivation of dynamics-aware reward augmentation (DARA) framework built on prior model-free and model-based formulations. DARA is simple and general, can accommodate various offline RL methods, and can be implemented in just a few lines of code on top of dataloader at training. In our offline dynamics adaptation setting, we also release a dataset, including the Gym-MuJoCo tasks (Walker2d, Hopper and HalfCheetah), with dynamics (mass, joint) shift compared to D4RL, and a 12-DoF quadruped robot in both simulator and real-world. With only modest amounts of target offline data, we show that DARA-based offline methods can acquire an adaptive policy for the target tasks and achieve better performance compared to baselines in both simulated and real-world tasks.

## 2    RELATED WORK

Offline RL describes the setting in which a learner has access to only a fixed dataset of experience, while no interactive data collection is allowed during policy learning (Levine et al., 2020). Prior work commonly *assumes* that the offline experience is collected by some behavior policies on the same environment that the learned policy be deployed on. Thus, the main difficulty of such offline setting is the state-action distribution shift (Fujimoto et al., 2019; Liu et al., 2018). Algorithms address this issue by following the two main directions: the *model-free* and *model-based* offline RL.

*Model-free* methods for such setting typically fall under three categories: 1) Typical methods mitigate this problem by explicitly (Fujimoto et al., 2019; Kumar et al., 2019; Wu et al., 2019) or implicitly (Siegel et al., 2020; Peng et al., 2019; Abdolmaleki et al., 2018) constraining the learned policy away from OOD state-action pairs. 2) Conservative estimation based methods learn pessimistic value functions to prevent the overestimation (Kumar et al., 2020a; Xu et al., 2021). 3) Importance sampling based methods directly estimate the state-marginal importance ratio and obtain an unbiased value estimation (Zhang et al., 2020; Nachum & Dai, 2020; Nachum et al., 2019b).

*Model-based* methods typically eliminate the state-action distribution shift by incorporating a reward penalty, which relies on the uncertainty quantification of the learned dynamics (Kidambi et al., 2020; Yu et al., 2020). To remove this uncertainty estimation, Yu et al. (2021) learns conservative critic function by penalizing the values of the generated state-action pairs that are not in the offline dataset.

These methods, however, define their objective based on the state-action distribution shift, and ignore the potential dynamics shift between the fixed offline data and the target MDP. In contrast, we account for dynamics (state-action-next-state) shift and explicitly propose the dynamics aware reward augmentation. A counterpart, close to our work, is off-dynamics RL (Eysenbach et al., 2021), where they set up dynamics shift in the interactive environment while we focus on the offline setting.

## 3 PRELIMINARIES

We study RL in the framework of Markov decision processes (MDPs) specified by the tuple $M :=$ $(\mathcal{S}, \mathcal{A}, r, T, \rho_0, \gamma)$, where $\mathcal{S}$ and $\mathcal{A}$ denote the state and action spaces, $r(\mathbf{s}, \mathbf{a}) \in [-R_{max}, R_{max}]$ is the reward function, $T(\mathbf{s}'|\mathbf{s}, \mathbf{a})$ is the transition dynamics, $\rho_0(\mathbf{s})$ is the initial state distribution, and $\gamma$ is the discount factor. The goal in RL is to optimize a policy $\pi(\mathbf{a}|\mathbf{s})$ that maximizes the expected discounted return $\eta_M(\pi) := \mathbb{E}_{\tau \sim p_M^\pi(\tau)} [\sum_{t=0}^\infty \gamma^t r(\mathbf{s}_t, \mathbf{a}_t)]$, where $\tau := (\mathbf{s}_0, \mathbf{a}_0, \mathbf{s}_1, \mathbf{a}_1, ...)$. We also define Q-values $Q(\mathbf{s}, \mathbf{a}) := \mathbb{E}_{\tau \sim p_M^\pi(\tau)} [\sum_{t=0}^\infty \gamma^t r(\mathbf{s}_t, \mathbf{a}_t)|\mathbf{s}_0 = \mathbf{s}, \mathbf{a}_0 = \mathbf{a}]$, V-values $V(\mathbf{s}) := \mathbb{E}_{\mathbf{a} \sim \pi(\mathbf{a}|\mathbf{s})} [Q(\mathbf{s}, \mathbf{a})]$, and the (unnormalized) state visitation distribution $d_M^\pi(\mathbf{s}) := \sum_{t=0}^\infty \gamma^t P(\mathbf{s}|\pi, M, t)$, where $P(\mathbf{s}|\pi, M, t)$ denotes the probability of reaching state $\mathbf{s}$ at time $t$ by running $\pi$ in $M$.

In the *offline RL* problem, we are provided with a static dataset $\mathcal{D} := \{(\mathbf{s}, \mathbf{a}, r, \mathbf{s}')\}$, which consists of transition tuples from trajectories collected by running one or more behavioral policies, denoted by $\pi_b$, on MDP $M$. With a slight abuse of notation, we write $\mathcal{D} = \{(\mathbf{s}, \mathbf{a}, r, \mathbf{s}') \sim d_\mathcal{D}(\mathbf{s})\pi_b(\mathbf{a}|\mathbf{s})r(\mathbf{s}, \mathbf{a})T(\mathbf{s}'|\mathbf{s}, \mathbf{a})\}$, where the $d_\mathcal{D}(\mathbf{s})$ denotes state-marginal distribution in $\mathcal{D}$. In the offline setting, the goal is typically to learn the best possible policy using the fixed offline dataset.

**Model-free RL** algorithms based on dynamic programming typically perform policy iteration to find the optimal policy. Such methods iteratively conduct 1) policy improvement with $\mathcal{G}_M Q :=$ $\arg\max_\pi \mathbb{E}_{\mathbf{s} \sim d_M^\pi(\mathbf{s}), \mathbf{a} \sim \pi(\mathbf{a}|\mathbf{s})} [Q(\mathbf{s}, \mathbf{a})]$ and 2) policy evaluation by iterating the Bellman equation $Q(\mathbf{s}, \mathbf{a}) = \mathcal{B}_M^\pi Q(\mathbf{s}, \mathbf{a}) := r(\mathbf{s}, \mathbf{a}) + \gamma \mathbb{E}_{\mathbf{s}' \sim T(\mathbf{s}'|\mathbf{s}, \mathbf{a}), \mathbf{a}' \sim \pi(\mathbf{a}'|\mathbf{s}')} [Q(\mathbf{s}', \mathbf{a}')]$ over $d_M^\pi(\mathbf{s})\pi(\mathbf{a}|\mathbf{s})$. Given off-policy $\mathcal{D}$, we resort to 1) improvement with $\mathcal{G}_\mathcal{D} Q := \arg\max_\pi \mathbb{E}_{\mathbf{s} \sim d_\mathcal{D}(\mathbf{s}), \mathbf{a} \sim \pi(\mathbf{a}|\mathbf{s})} [Q(\mathbf{s}, \mathbf{a})]$ and 2) evaluation by iterating $Q(\mathbf{s}, \mathbf{a}) = \mathcal{B}_\mathcal{D}^\pi Q(\mathbf{s}, \mathbf{a}) := r(\mathbf{s}, \mathbf{a}) + \gamma \mathbb{E}_{\mathbf{s}' \sim T_\mathcal{D}(\mathbf{s}'|\mathbf{s}, \mathbf{a}), \mathbf{a}' \sim \pi(\mathbf{a}'|\mathbf{s}')} [Q(\mathbf{s}', \mathbf{a}')]$ over all $(\mathbf{s}, \mathbf{a})$ in $\mathcal{D}$. Specifically, given any initial $Q^0$, it iterates[1]

$$\text{Policy improvement: } \pi^{k+1} = \mathcal{G}_\mathcal{D} Q^k, \qquad \text{Policy evaluation: } Q^{k+1} = \mathcal{B}_\mathcal{D}^{\pi^{k+1}} Q^k. \tag{1}$$

**Model-free offline RL** based on the above iteration suffers from the state-action distribution shift, *i.e.*, policy evaluation $\mathcal{B}_\mathcal{D}^{\pi^k} Q^{k-1}$ may encounter unfamiliar state action regime that is not covered by the fixed offline dataset $\mathcal{D}$, causing erroneous estimation of $Q^k$. Policy improvement $\mathcal{G}_\mathcal{D} Q^k$ further exaggerates such error, biasing policy $\pi^{k+1}$ towards out-of-distribution (OOD) actions with erroneously high Q-values. To address this distribution shift, prior works 1) explicitly constrain policy to be close to the behavior policy (Fujimoto et al., 2019; Kumar et al., 2019; Wu et al., 2019; Ghasemipour et al., 2021), introducing penalty $\alpha D(\pi(\mathbf{a}|\mathbf{s}), \pi_b(\mathbf{a}|\mathbf{s}))$ into $\mathcal{G}_\mathcal{D}$ or $\mathcal{B}_\mathcal{D}^\pi$ in Equation 1:

$$
\begin{aligned}
\mathcal{G}_\mathcal{D} Q &= \arg\max_\pi \mathbb{E}_{\mathbf{s} \sim d_\mathcal{D}(\mathbf{s}), \mathbf{a} \sim \pi(\mathbf{a}|\mathbf{s})} [Q(\mathbf{s}, \mathbf{a}) - \alpha D(\pi(\mathbf{a}|\mathbf{s}), \pi_b(\mathbf{a}|\mathbf{s}))], \\
\mathcal{B}_\mathcal{D}^\pi Q(\mathbf{s}, \mathbf{a}) &= r(\mathbf{s}, \mathbf{a}) + \gamma \mathbb{E}_{\mathbf{s}' \sim T_\mathcal{D}(\mathbf{s}'|\mathbf{s}, \mathbf{a}), \mathbf{a}' \sim \pi(\mathbf{a}'|\mathbf{s}')} [Q(\mathbf{s}', \mathbf{a}') - \alpha D(\pi(\mathbf{a}'|\mathbf{s}'), \pi_b(\mathbf{a}'|\mathbf{s}'))],
\end{aligned}
\tag{2}
$$

where $D$ is a divergence function between distributions over actions (*e.g.*, MMD or KL divergence), or 2) train pessimistic value functions (Kumar et al., 2020a; Yu et al., 2021; Xu et al., 2021), penalizing Q-values at states in the offline dataset $\mathcal{D}$ for actions generated by the current policy $\pi$:

$$Q = \arg\min_Q \mathbb{E}_{\mathbf{s} \sim d_\mathcal{D}(\mathbf{s}), \mathbf{a} \sim \pi(\mathbf{a}|\mathbf{s})} [Q(\mathbf{s}, \mathbf{a})], \quad \text{s.t. } Q = \mathcal{B}_\mathcal{D}^\pi Q. \tag{3}$$

**Model-based RL** algorithms iteratively 1) model the transition dynamics $T(\mathbf{s}'|\mathbf{s}, \mathbf{a})$, using the data collected in $M$: $\max_{\hat{T}} \mathbb{E}_{\mathbf{s}, \mathbf{a}, \mathbf{s}' \sim d_M^\pi(\mathbf{s})\pi(\mathbf{a}|\mathbf{s})T(\mathbf{s}'|\mathbf{s}, \mathbf{a})} [\log \hat{T}(\mathbf{s}'|\mathbf{s}, \mathbf{a})]$, and 2) infer a policy $\pi$ from the modeled $\hat{M} = (\mathcal{S}, \mathcal{A}, r, \hat{T}, \rho_0, \gamma)$, where we assume that $r$ and $\rho_0$ are known, maximizing $\eta_{\hat{M}}(\pi)$ with a planner or the Dyna-style algorithms (Sutton, 1990). In this paper, we focus on the latter.

**Model-based offline RL** algorithms similarly suffer from OOD state-action (Kidambi et al., 2020; Cang et al., 2021) if we directly apply policy iteration over $\hat{T} := \max_{\hat{T}} \mathbb{E}_{\mathbf{s}, \mathbf{a}, \mathbf{s}' \sim \mathcal{D}} [\log \hat{T}(\mathbf{s}'|\mathbf{s}, \mathbf{a})]$. Like the conservative estimation approach described in Equation 3, recent conservative model-based offline RL methods provide the policy with a penalty for visiting states under the estimated $\hat{T}$ where $\hat{T}$ is likely to be incorrect. Taking $u(\mathbf{s}, \mathbf{a})$ as the oracle uncertainty (Yu et al., 2020) that provides a consistent estimate of the accuracy of model $\hat{T}$ at $(\mathbf{s}, \mathbf{a})$, we can modify the reward function to obtain a conservative MDP: $\hat{M}_c = (\mathcal{S}, \mathcal{A}, r - \alpha u, \hat{T}, \rho_0, \gamma)$, then learn a policy $\pi$ by maximizing $\eta_{\hat{M}_c}(\pi)$.

---

[1]For parametric Q-function, we often perform $Q^{k+1} \leftarrow \arg\min_Q \mathbb{E}_{(\mathbf{s}, \mathbf{a}) \sim \mathcal{D}} [(\mathcal{B}_\mathcal{D}^{\pi^{k+1}} Q^k(\mathbf{s}, \mathbf{a}) - Q(\mathbf{s}, \mathbf{a}))^2]$.

## 4 PROBLEM FORMULATION

In standard offline RL problem, the static offline dataset $\mathcal{D}$ consists of samples $\{(\mathbf{s}, \mathbf{a}, r, \mathbf{s}') \sim d_{\mathcal{D}}(\mathbf{s})\pi_b(\mathbf{a}|\mathbf{s})r(\mathbf{s}, \mathbf{a})T(\mathbf{s}'|\mathbf{s}, \mathbf{a})\}$. Although offline RL methods learn policy for the target MDP $M := (\mathcal{S}, \mathcal{A}, r, T, \rho_0, \gamma)$ without (costly) online data, as we shown in Figure 1, it requires a fair amount of (target) offline data $\mathcal{D}$ collected on $M$. Suppose we have another (source) offline dataset $\mathcal{D}'$, consisting of samples $\{(\mathbf{s}, \mathbf{a}, r, \mathbf{s}') \sim d_{\mathcal{D}'}(\mathbf{s})\pi_{b'}(\mathbf{a}|\mathbf{s})r(\mathbf{s}, \mathbf{a})T'(\mathbf{s}'|\mathbf{s}, \mathbf{a})\}$ collected by the behavior policy $\pi_{b'}$ on MDP $M' := (\mathcal{S}, \mathcal{A}, r, T', \rho_0, \gamma)$, then we hope the transfer of knowledge between offline dataset $\{\mathcal{D}' \cup \mathcal{D}\}$ can reduce the data requirements on $\mathcal{D}$ for learning policy for the target $M$.

### 4.1 DYNAMICS SHIFT IN OFFLINE RL

Although offline RL methods in Section 3 have incorporated the state-action distribution constrained backups (policy constraints or conservative estimation), they also fail to learn an adaptive policy for the target MDP $M$ with the mixed datasets $\{\mathcal{D}' \cup \mathcal{D}\}$, as we show in Figure 4 (Appendix). We attribute this failure to the dynamics shift (Definition 2) between $\mathcal{D}'$ and $M$ in this adaptation setting.

**Definition 1** *(Empirical MDP) An empirical MDP estimated from $\mathcal{D}$ is $\hat{M} := (\mathcal{S}, \mathcal{A}, r, \hat{T}, \rho_0, \gamma)$ where $\hat{T} = \max_{\hat{T}} \mathbb{E}_{\mathbf{s}, \mathbf{a}, \mathbf{s}' \sim \mathcal{D}}[\log \hat{T}(\mathbf{s}'|\mathbf{s}, \mathbf{a})]$ and $\hat{T}(\mathbf{s}'|\mathbf{s}, \mathbf{a}) = 0$ for all $(\mathbf{s}, \mathbf{a}, \mathbf{s}')$ not in dataset $\mathcal{D}$.*

**Definition 2** *(Dynamics shift) Let $\hat{M} := (\mathcal{S}, \mathcal{A}, r, \hat{T}, \rho_0, \gamma)$ be the empirical MDP estimated from $\mathcal{D}$. To evaluate a policy $\pi$ for $M := (\mathcal{S}, \mathcal{A}, r, T, \rho_0, \gamma)$ with offline dataset $\mathcal{D}$, we say that the dynamics shift (between $\mathcal{D}$ and $M$) in offline RL happens if there exists at least one transition pair $(\mathbf{s}, \mathbf{a}, \mathbf{s}') \in \{(\mathbf{s}, \mathbf{a}, \mathbf{s}') : d_{\hat{M}}^{\pi}(\mathbf{s})\pi(\mathbf{a}|\mathbf{s})\hat{T}(\mathbf{s}'|\mathbf{s}, \mathbf{a}) > 0\}$ such that $\hat{T}(\mathbf{s}'|\mathbf{s}, \mathbf{a}) \neq T(\mathbf{s}'|\mathbf{s}, \mathbf{a})$.*

In practice, for a stochastic $M$ and any finite offline data $\mathcal{D}$ collected in $M$, there always exists the dynamics shift. The main concern is that finite samples are always not sufficient to exactly model stochastic dynamics. Following Fujimoto et al. (2019), we thus assume both MDPs $M$ and $M'$ are deterministic, which means the empirical $\hat{M}$ and $\hat{M}'$ are both also deterministic. More importantly, such assumption enables us to explicitly characterize the dynamics shift under finite offline samples.

**Lemma 1** *Under deterministic transition dynamics, there is no dynamics shift between $\mathcal{D}$ and $M$.*

For offline RL tasks, prior methods generally apply $\mathcal{B}_{\mathcal{D}}^{\pi}Q$ along with the state-action distribution correction (Equations 2 and 3), which overlooks the potential dynamics shift between the (source) offline dataset and the target MDP (*e.g.*, $\mathcal{D}' \rightarrow M$). As a result, these methods do not scale well to the setting in which dynamics shift happens, *e.g.*, learning an adaptive policy for $M$ with (source) $\mathcal{D}'$.

### 4.2 DYNAMICS SHIFT IN MODEL-FREE AND MODEL-BASED OFFLINE FORMULATIONS

From the **model-free** (policy iteration) view, an exact policy evaluation on $M$ is characterized by iterating $Q(\mathbf{s}, \mathbf{a}) = \mathcal{B}_M^{\pi}Q(\mathbf{s}, \mathbf{a})$ for all $(\mathbf{s}, \mathbf{a})$ such that $d_M^{\pi}(\mathbf{s})\pi(\mathbf{a}|\mathbf{s}) > 0$. Thus, to formalize the policy evaluation with offline $\mathcal{D}$ or $\mathcal{D}'$ (for an adaptive $\pi$ on target $M$), we require that Bellman operator $\mathcal{B}_{\mathcal{D}}^{\pi}Q(\mathbf{s}, \mathbf{a})$ or $\mathcal{B}_{\mathcal{D}'}^{\pi}Q(\mathbf{s}, \mathbf{a})$ approximates the oracle $\mathcal{B}_M^{\pi}Q(\mathbf{s}, \mathbf{a})$ for all $(\mathbf{s}, \mathbf{a})$ in $S_{\pi}$ or $S_{\pi}'$, where $S_{\pi}$ and $S_{\pi}'$ denote the sets $\{(\mathbf{s}, \mathbf{a}) : d_{\mathcal{D}}(\mathbf{s})\pi(\mathbf{a}|\mathbf{s}) > 0\}$ and $\{(\mathbf{s}, \mathbf{a}) : d_{\mathcal{D}'}(\mathbf{s})\pi(\mathbf{a}|\mathbf{s}) > 0\}$ respectively.

1) To evaluate a policy $\pi$ for $M$ with $\mathcal{D}$ (*i.e.*, calling the Bellman operator $\mathcal{B}_{\mathcal{D}}^{\pi}$), notable model-free offline method BCQ (Fujimoto et al., 2019) translates the requirement of $\mathcal{B}_{\mathcal{D}}^{\pi} = \mathcal{B}_M^{\pi}$ into the requirement of $\hat{T}(\mathbf{s}'|\mathbf{s}, \mathbf{a}) = T(\mathbf{s}'|\mathbf{s}, \mathbf{a})$. Note that under deterministic environments, we have the property that for all $(\mathbf{s}, \mathbf{a}, \mathbf{s}')$ in offline data $\mathcal{D}$, $\hat{T}(\mathbf{s}'|\mathbf{s}, \mathbf{a}) = T(\mathbf{s}'|\mathbf{s}, \mathbf{a})$ (Lemma 1). As a result, such property permits BCQ to evaluate a policy $\pi$ by calling $\mathcal{B}_{\mathcal{D}}^{\pi}$, replacing the oracle $\mathcal{B}_M^{\pi}$, meanwhile constraining $S_{\pi}$ to be a subset of the support of $d_{\mathcal{D}}(\mathbf{s})\pi_b(\mathbf{a}|\mathbf{s})$. This means a policy $\pi$ which only traverses transitions contained in (target) offline data $\mathcal{D}$, can be evaluated on $M$ without error.

2) To evaluate a policy $\pi$ for $M$ with $\mathcal{D}'$ (*i.e.*, calling the Bellman operator $\mathcal{B}_{\mathcal{D}'}^{\pi}$), we have lemma 2:

**Lemma 2** *Dynamics shift produces that $\mathcal{B}_{\mathcal{D}'}^{\pi}Q(\mathbf{s}, \mathbf{a}) \neq \mathcal{B}_M^{\pi}Q(\mathbf{s}, \mathbf{a})$ for some $(\mathbf{s}, \mathbf{a})$ in $S_{\pi}'$.*

With the offline data $\mathcal{D}'$, lemma 2 suggests that the above requirement $\mathcal{B}_{\mathcal{D}'}^{\pi} = \mathcal{B}_M^{\pi}$ becomes infeasible, which limits the practical applicability of prior offline RL methods under the dynamics shift.

To be specific, characterizing an adaptive policy for target MDP $M$ with $\mathcal{D}'$ moves beyond the reach of the off-policy evaluation based on iterating $Q = \mathcal{B}^\pi_{\mathcal{D}'} Q$ (Equations 2 and 3). Such iteration may cause the evaluated $Q$ (or learned policy $\pi$) overfits to $\hat{T}'$ and struggle to adapt to the target $T$. To overcome the dynamics shift, we would like to resort an additional compensation $\Delta_{\hat{T}', T}$ such that

$$\mathcal{B}^\pi_{\mathcal{D}'} Q(\mathbf{s}, \mathbf{a}) + \Delta_{\hat{T}', T}(\mathbf{s}, \mathbf{a}) = \mathcal{B}^\pi_M Q(\mathbf{s}, \mathbf{a}) \tag{4}$$

for all $(\mathbf{s}, \mathbf{a})$ in $S'_\pi$. Thus, we can apply $\mathcal{B}^\pi_{\mathcal{D}'} Q + \Delta_{\hat{T}', T}$ to act as a substitute for the oracle $\mathcal{B}^\pi_M Q$.

From the **model-based** view, the oracle $\eta_M(\pi)$ (calling the Bellman operator $\mathcal{B}^\pi_M$ on the target $M$) and the viable $\eta_{\hat{M}'}(\pi)$ (calling $\mathcal{B}^\pi_{\hat{M}'}$ on the estimated $\hat{M}'$ from source $\mathcal{D}'$) have the following lemma.

**Lemma 3** *Let $\mathcal{B}^\pi_M V(\mathbf{s}) = \mathbb{E}_{\mathbf{a} \sim \pi(\mathbf{a}|\mathbf{s})} \left[ r(\mathbf{s}, \mathbf{a}) + \gamma \mathbb{E}_{\mathbf{s}' \sim T(\mathbf{s}'|\mathbf{s}, \mathbf{a})} [V(\mathbf{s}')] \right]$. For any $\pi$, we have:*

$$\eta_{\hat{M}'}(\pi) = \eta_M(\pi) + \mathbb{E}_{\mathbf{s} \sim d^\pi_{\hat{M}'}(\mathbf{s})} \left[ \mathcal{B}^\pi_{\hat{M}'} V_M(\mathbf{s}) - \mathcal{B}^\pi_M V_M(\mathbf{s}) \right].$$

Lemma 3 states that if we maximize $\eta_{\hat{M}'}(\pi)$ subject to $|\mathbb{E}_{\mathbf{s} \sim d^\pi_{\hat{M}'}(\mathbf{s})} [\mathcal{B}^\pi_{\hat{M}'} V_M(\mathbf{s}) - \mathcal{B}^\pi_M V_M(\mathbf{s})]| \leq \epsilon$, $\eta_M(\pi)$ will be improved. If $\mathcal{F}$ is a set of functions $f : \mathcal{S} \to \mathbb{R}$ that contains $V_M$, then we have

$$\left| \mathbb{E}_{\mathbf{s} \sim d^\pi_{\hat{M}'}(\mathbf{s})} \left[ \mathcal{B}^\pi_{\hat{M}'} V_M(\mathbf{s}) - \mathcal{B}^\pi_M V_M(\mathbf{s}) \right] \right| \leq \gamma \mathbb{E}_{\mathbf{s}, \mathbf{a} \sim d^\pi_{\hat{M}'}(\mathbf{s}) \pi(\mathbf{a}|\mathbf{s})} \left[ d_\mathcal{F}(\hat{T}'(\mathbf{s}'|\mathbf{s}, \mathbf{a}), T(\mathbf{s}'|\mathbf{s}, \mathbf{a})) \right], \quad (5)$$

where $d_\mathcal{F}(\hat{T}'(\mathbf{s}'|\mathbf{s}, \mathbf{a}), T(\mathbf{s}'|\mathbf{s}, \mathbf{a})) = \sup_{f \in \mathcal{F}} |\mathbb{E}_{\mathbf{s}' \sim \hat{T}'(\mathbf{s}'|\mathbf{s}, \mathbf{a})} [f(\mathbf{s}')] - \mathbb{E}_{\mathbf{s}' \sim T(\mathbf{s}'|\mathbf{s}, \mathbf{a})} [f(\mathbf{s}')]|$, which is the integral probability metric (IPM). Note that if we directly follow the *admissible error* assumption in MOPO (Yu et al., 2020) *i.e.*, assuming $d_\mathcal{F}(\hat{T}'(\mathbf{s}'|\mathbf{s}, \mathbf{a}), T(\mathbf{s}'|\mathbf{s}, \mathbf{a})) \leq u(\mathbf{s}, \mathbf{a})$ for all $(\mathbf{s}, \mathbf{a})$, this would be too restrictive: given that $\hat{T}'$ is estimated from the source offline samples collected under $T'$, not the target $T$, thus such error would not decrease as the source data increases. Further, we find

$$d_\mathcal{F}(\hat{T}'(\mathbf{s}'|\mathbf{s}, \mathbf{a}), T(\mathbf{s}'|\mathbf{s}, \mathbf{a})) \leq d_\mathcal{F}(\hat{T}'(\mathbf{s}'|\mathbf{s}, \mathbf{a}), \hat{T}(\mathbf{s}'|\mathbf{s}, \mathbf{a})) + d_\mathcal{F}(\hat{T}(\mathbf{s}'|\mathbf{s}, \mathbf{a}), T(\mathbf{s}'|\mathbf{s}, \mathbf{a})). \tag{6}$$

Thus, we can bound the $d_\mathcal{F}(\hat{T}', T)$ term with the admissible error assumption over $d_\mathcal{F}(\hat{T}, T)$, as in MOPO, and the auxiliary constraints $d_\mathcal{F}(\hat{T}', \hat{T})$. See next section for the detailed implementation.

**In summary**, we show that both prior offline model-free and model-based formulations suffer from the dynamics shift, which also suggests us to learn a modification ($\Delta$ or $d_\mathcal{F}$) to eliminate this shift.

## 5 DYNAMICS-AWARE REWARD AUGMENTATION

In this section, we propose the dynamics-aware reward augmentation (DARA), a simple data augmentation procedure based on prior (model-free and model-based) offline RL methods. We first provide an overview of our offline reward augmentation motivated by the compensation $\Delta_{\hat{T}', T}$ in Equation 4 and the auxiliary constraints $d_\mathcal{F}(\hat{T}', \hat{T})$ in Equation 6, and then describe its theoretical derivation in both model-free and model-based formulations. With the (reduced) target offline data $\mathcal{D}$ and the source offline data $\mathcal{D}'$, we summarize the overall DARA framework in Algorithm 1.

---

**Algorithm 1** Framework for Dynamics-Aware Reward Augmentation (DARA)

---

**Require:** Target offline data $\mathcal{D}$ (reduced) and source offline data $\mathcal{D}'$
1: Learn classifiers ($q_{\text{sas}}$ and $q_{\text{sa}}$) that distinguish source data $\mathcal{D}'$ from target data $\mathcal{D}$. (See Appendix A.1.3)
2: Set dynamics-aware $\Delta r(\mathbf{s}_t, \mathbf{a}_t, \mathbf{s}_{t+1}) = \log \frac{q_{\text{sas}}(\text{source}|\mathbf{s}_t, \mathbf{a}_t, \mathbf{s}_{t+1})}{q_{\text{sas}}(\text{target}|\mathbf{s}_t, \mathbf{a}_t, \mathbf{s}_{t+1})} - \log \frac{q_{\text{sa}}(\text{source}|\mathbf{s}_t, \mathbf{a}_t)}{q_{\text{sa}}(\text{target}|\mathbf{s}_t, \mathbf{a}_t)}$.
3: Modify rewards for all $(\mathbf{s}_t, \mathbf{a}_t, r_t, \mathbf{s}_{t+1})$ in $\mathcal{D}'$: $r_t \leftarrow r_t - \eta \Delta r$.
4: Learn policy with $\{\mathcal{D} \cup \mathcal{D}'\}$ using prior model-free or model-based offline RL algorithms.

---

### 5.1 DYNAMICS-AWARE REWARD AUGMENTATION IN MODEL-FREE FORMULATION

Motivated by the well established connection of RL and probabilistic inference (Levine, 2018), we first cast the model-free RL problem as that of inference in a particular probabilistic model. Specifically, we introduce the binary random variable $\mathcal{O}$ that denotes whether the trajectory $\tau :=$

$(\mathbf{s}_0, \mathbf{a}_0, \mathbf{s}_1, ...)$ is optimal ($\mathcal{O} = 1$) or not ($\mathcal{O} = 0$). The likelihood of a trajectory can then be modeled as $p(\mathcal{O} = 1|\tau) = \exp\left(\sum_t r_t/\eta\right)$, where $r_t := r(\mathbf{s}_t, \mathbf{a}_t)$ and $\eta > 0$ is a temperature parameter.

**(Reward Augmentation with Explicit Policy/Value Constraints)** We now introduce a variational distribution $p_{\hat{M}'}^\pi(\tau) = p(\mathbf{s}_0) \prod_{t=1} \hat{T}'(\mathbf{s}_{t+1}|\mathbf{s}_t, \mathbf{a}_t)\pi(\mathbf{a}_t|\mathbf{s}_t)$ to approximate the posterior distribution $p_M^\pi(\tau|\mathcal{O} = 1)$, which leads to the evidence lower bound of $\log p_M^\pi(\mathcal{O} = 1)$:

$$\log p_M^\pi(\mathcal{O} = 1) = \log \mathbb{E}_{\tau \sim p_M^\pi(\tau)}\left[p(\mathcal{O} = 1|\tau)\right] \geq \mathbb{E}_{\tau \sim p_{\hat{M}'}^\pi(\tau)}\left[\log p(\mathcal{O} = 1|\tau) + \log \frac{p_M^\pi(\tau)}{p_{\hat{M}'}^\pi(\tau)}\right]$$

$$= \mathbb{E}_{\tau \sim p_{\hat{M}'}^\pi(\tau)}\left[\sum_t \left(r_t/\eta - \log \frac{\hat{T}'(\mathbf{s}_{t+1}|\mathbf{s}_t, \mathbf{a}_t)}{T(\mathbf{s}_{t+1}|\mathbf{s}_t, \mathbf{a}_t)}\right)\right]. \tag{7}$$

Since we are interested in infinite horizon problems, we introduce the discount factor $\gamma$ and take the limit of steps in each rollout, *i.e.*, $H \to \infty$. Thus, the RL problem on the MDP $M$, cast as the inference problem $\arg\max_\pi \log p_M^\pi(\mathcal{O} = 1)$, can be stated as a maximum of the lower bound $\mathbb{E}_{\tau \sim p_{\hat{M}'}^\pi(\tau)}\left[\sum_{t=0}^\infty \gamma^t \left(r_t - \eta \log \frac{\hat{T}'(\mathbf{s}_{t+1}|\mathbf{s}_t, \mathbf{a}_t)}{T(\mathbf{s}_{t+1}|\mathbf{s}_t, \mathbf{a}_t)}\right)\right]$. This is equivalent to an RL problem on $\hat{M}'$ with the augmented reward $r \leftarrow r(\mathbf{s}, \mathbf{a}) - \eta \log \frac{\hat{T}'(\mathbf{s}'|\mathbf{s}, \mathbf{a})}{T(\mathbf{s}'|\mathbf{s}, \mathbf{a})}$. Intuitively, the $-\eta \log \frac{\hat{T}'(\mathbf{s}'|\mathbf{s}, \mathbf{a})}{T(\mathbf{s}'|\mathbf{s}, \mathbf{a})}$ term discourages transitions (state-action-next-state) in $\mathcal{D}'$ that have low transition probability in the target $M$. In the model-free offline setting, we can add the explicit policy or Q-value constraints (Equations-2 and 3) to mitigate the OOD state-actions. Thus, such formulation allows the oracle $\mathcal{B}_M^\pi$ to be re-expressed by $\mathcal{B}_{\mathcal{D}'}^\pi$ and the modification $\log \frac{\hat{T}'}{T}$, which makes the motivation in Equation 4 practical.

**(Reward Augmentation with Implicit Policy Constraints)** If we introduce the variational distribution $p_{\hat{M}'}^{\pi'}(\tau) := p(\mathbf{s}_0) \prod_{t=1} \hat{T}'(\mathbf{s}_{t+1}|\mathbf{s}_t, \mathbf{a}_t)\pi'(\mathbf{a}_t|\mathbf{s}_t)$, we can recover the weighted-regression-style (Wang et al., 2020; Peng et al., 2019; Abdolmaleki et al., 2018; Peters et al., 2010) objective by maximizing $\mathcal{J}(\pi', \pi) := \mathbb{E}_{\tau \sim p_{\hat{M}'}^{\pi'}(\tau)}\left[\sum_{t=0}^\infty \gamma^t \left(r_t - \eta \log \frac{\hat{T}'(\mathbf{s}_{t+1}|\mathbf{s}_t, \mathbf{a}_t)}{T(\mathbf{s}_{t+1}|\mathbf{s}_t, \mathbf{a}_t)} - \eta \log \frac{\pi'(\mathbf{a}_t|\mathbf{s}_t)}{\pi(\mathbf{a}_t|\mathbf{s}_t)}\right)\right]$ (lower bound of $\log p_M^\pi(\mathcal{O} = 1)$). Following the Expectation Maximization (EM) algorithm, we can maximize $\mathcal{J}(\pi', \pi)$ by iteratively (E-step) improving $\mathcal{J}(\pi', \cdot)$ w.r.t. $\pi'$ and (M-step) updating $\pi$ w.r.t. $\pi'$.

*(E-step)* We define $\tilde{Q}(\mathbf{s}, \mathbf{a}, \mathbf{s}') = \mathbb{E}_{\tau \sim p_{\hat{M}'}^{\pi'}(\tau)}\left[\sum_t \gamma^t \log \frac{\hat{T}'(\mathbf{s}'|\mathbf{s}, \mathbf{a})}{T(\mathbf{s}'|\mathbf{s}, \mathbf{a})}|\mathbf{s}_0 = \mathbf{s}, \mathbf{a}_0 = \mathbf{a}, \mathbf{s}_1 = \mathbf{s}'\right]$. Then, given offline data $\mathcal{D}'$, we can rewrite $\mathcal{J}(\pi', \cdot)$ as a constrained objective (Abdolmaleki et al., 2018):

$$\max_{\pi'} \mathbb{E}_{d_{\mathcal{D}'}(\mathbf{s})\pi'(\mathbf{a}|\mathbf{s})\hat{T}'(\mathbf{s}'|\mathbf{s}, \mathbf{a})}\left[Q(\mathbf{s}, \mathbf{a}) - \eta\tilde{Q}(\mathbf{s}, \mathbf{a}, \mathbf{s}')\right], \quad \text{s.t.} \ \mathbb{E}_{\mathbf{s} \sim d_{\mathcal{D}'}(\mathbf{s})}\left[D_{\mathrm{KL}}(\pi'(\mathbf{a}|\mathbf{s})\|\pi(\mathbf{a}|\mathbf{s}))\right] \leq \epsilon.$$

When considering a fixed $\pi$, the above optimization over $\pi'$ can be solved analytically (Vieillard et al., 2020; Geist et al., 2019; Peng et al., 2019). The optimal $\pi'_*$ is then given by $\pi'_*(\mathbf{a}|\mathbf{s}) \propto \pi(\mathbf{a}|\mathbf{s}) \exp(Q(\mathbf{s}, \mathbf{a})) \exp(-\eta\tilde{Q}(\mathbf{s}, \mathbf{a}, \hat{T}'(\mathbf{s}'|\mathbf{s}, \mathbf{a})))$. As the policy evaluation in Equation 1 (Footnote-2), we estimate $Q(\mathbf{s}, \mathbf{a})$ and $\tilde{Q}(\mathbf{s}, \mathbf{a}, \mathbf{s}')$ by minimizing the Bellman error with offline samples in $\mathcal{D}'$.

*(M-step)* Then, we can project $\pi'_*$ onto the manifold of the parameterized $\pi$:

$$\arg\min_\pi \ \mathbb{E}_{\mathbf{s} \sim d_{\mathcal{D}'}(\mathbf{s})}\left[D_{\mathrm{KL}}(\pi'_*(\mathbf{a}|\mathbf{s})\|\pi(\mathbf{a}|\mathbf{s}))\right]$$

$$= \arg\max_\pi \ \mathbb{E}_{\mathbf{s}, \mathbf{a}, \mathbf{s}' \sim \mathcal{D}'}\left[\log \pi(\mathbf{a}|\mathbf{s}) \exp(Q(\mathbf{s}, \mathbf{a})) \exp\left(-\eta\tilde{Q}(\mathbf{s}, \mathbf{a}, \mathbf{s}')\right)\right]. \tag{8}$$

From the regression view, prior work MPO (Abdolmaleki et al., 2018) infers actions with Q-value weighted regression, progressive approach compared to behavior cloning; however, such paradigm lacks the ability to capture transition dynamics. We explicitly introduce the $\exp(-\eta\tilde{Q}(\mathbf{s}, \mathbf{a}, \mathbf{s}'))$ term, which as we show in experiments, is a crucial component for eliminating the dynamics shift.

*Implementation:* In practice, we adopt offline samples in $\mathcal{D}$ to approximate the true dynamics $T$ of $M$, and introduce a pair of binary classifiers, $q_{\mathrm{sas}}(\cdot|\mathbf{s}, \mathbf{a}, \mathbf{s}')$ and $q_{\mathrm{sa}}(\cdot|\mathbf{s}, \mathbf{a})$, to replace $\log \frac{\hat{T}'(\mathbf{s}'|\mathbf{s}, \mathbf{a})}{T(\mathbf{s}'|\mathbf{s}, \mathbf{a})}$ as in Eysenbach et al. (2021): $\log \frac{\hat{T}'(\mathbf{s}'|\mathbf{s}, \mathbf{a})}{T(\mathbf{s}'|\mathbf{s}, \mathbf{a})} = \log \frac{q_{\mathrm{sas}}(\mathrm{source}|\mathbf{s}, \mathbf{a}, \mathbf{s}')}{q_{\mathrm{sas}}(\mathrm{target}|\mathbf{s}, \mathbf{a}, \mathbf{s}')} - \log \frac{q_{\mathrm{sa}}(\mathrm{source}|\mathbf{s}, \mathbf{a})}{q_{\mathrm{sa}}(\mathrm{target}|\mathbf{s}, \mathbf{a})}$. (See Appendix-A.1.3 for details). Although the amount of data $\mathcal{D}$ sampled from the target $M$ is reduced in our problem setup, we experimentally find that such classifiers are sufficient to achieve good performance.

## 5.2 DYNAMICS-AWARE REWARD AUGMENTATION IN MODEL-BASED FORMULATION

Following Equation 6, we then characterize the dynamics shift compensation term as in the above model-free analysis in the model-based offline formulation. We will find that across different derivations, our reward augmentation $\Delta r$ has always maintained the functional consistency and simplicity.

Following MOPO, we assume $\mathcal{F} = \{f : \|f\|_\infty \leq 1\}$, then we have $d_\mathcal{F}(\hat{T}'(\mathbf{s}'|\mathbf{s},\mathbf{a}), \hat{T}(\mathbf{s}'|\mathbf{s},\mathbf{a})) = D_{\mathrm{TV}}(\hat{T}'(\mathbf{s}'|\mathbf{s},\mathbf{a}), \hat{T}(\mathbf{s}'|\mathbf{s},\mathbf{a})) \leq (D_{\mathrm{KL}}(\hat{T}'(\mathbf{s}'|\mathbf{s},\mathbf{a}), \hat{T}(\mathbf{s}'|\mathbf{s},\mathbf{a}))/2)^{\frac{1}{2}}$, where $D_{\mathrm{TV}}$ is the total variance distance. Then we introduce the admissible error $u(\mathbf{s},\mathbf{a})$ such that $d_\mathcal{F}(\hat{T}(\mathbf{s}'|\mathbf{s},\mathbf{a}), T(\mathbf{s}'|\mathbf{s},\mathbf{a})) \leq u(\mathbf{s},\mathbf{a})$ for all $(\mathbf{s},\mathbf{a})$, and $\eta$ and $\delta$ such that $(D_{\mathrm{KL}}(\hat{T}', \hat{T})/2)^{\frac{1}{2}} \leq \eta D_{\mathrm{KL}}(\hat{T}', \hat{T}) + \delta$. Following Lemma 3, we thus can maximize the following lower bound with the samples in $\hat{M}'$ ($\lambda := \frac{\gamma R_{max}}{1-\gamma}$):

$$\eta_M(\pi) \geq \mathbb{E}_{\mathbf{s},\mathbf{a},\mathbf{s}' \sim d^\pi_{\hat{M}'}(\mathbf{s})\pi(\mathbf{a}|\mathbf{s})\hat{T}'(\mathbf{s}'|\mathbf{s},\mathbf{a})} \left[ r(\mathbf{s},\mathbf{a}) - \eta\lambda \log \frac{\hat{T}'(\mathbf{s}'|\mathbf{s},\mathbf{a})}{\hat{T}(\mathbf{s}'|\mathbf{s},\mathbf{a})} - \lambda u(\mathbf{s},\mathbf{a}) - \lambda\delta \right]. \quad (9)$$

*Implementation:* We model the dynamics $\hat{T}'$ and $\hat{T}$ with an ensemble of 2*N parameterized Gaussian distributions: $\mathcal{N}^i_{\hat{T}'}(\mu_{\theta'}(\mathbf{s},\mathbf{a}), \Sigma_{\phi'}(\mathbf{s},\mathbf{a}))$ and $\mathcal{N}^i_{\hat{T}}(\mu_\theta(\mathbf{s},\mathbf{a}), \Sigma_\phi(\mathbf{s},\mathbf{a}))$, where $i \in [1, N]$. We approximate $u$ with the maximum standard deviation of the learned models in the ensemble: $u(\mathbf{s},\mathbf{a}) = \max_{i=1}^N \|\Sigma_\phi(\mathbf{s},\mathbf{a})\|_{\mathrm{F}}$, omit the training-independent $\delta$, and treat $\lambda$ as a hyperparameter as in MOPO. For the $\log \frac{\hat{T}'}{\hat{T}}$ term, we resort to the above classifiers ($q_{\mathrm{sas}}$ and $q_{\mathrm{sa}}$) in model-free setting. (See Appendix-A.3.2 for comparison between using classifiers and estimated-dynamics ratio.)

## 6 EXPERIMENTS

We present empirical demonstrations of our dynamics-aware reward augmentation (DARA) in a variety of settings. We start with two simple control experiments that illustrate the significance of DARA under the domain (dynamics) adaptation setting. Then we incorporate DARA into state-of-the-art (model-free and model-based) offline RL methods and evaluate the performance on the D4RL tasks. Finally, we compare our framework to several cross-domain-based baselines on simulated and real-world tasks. Note that for the dynamics adaptation, we also release a (source) dataset as a complement to D4RL, along with the quadruped robot dataset in simulator (source) and real (target).

### 6.1 HOW DOES DARA HANDLE THE DYNAMICS SHIFT IN OFFLINE SETTING?

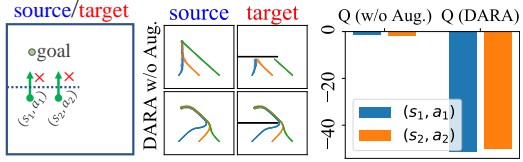
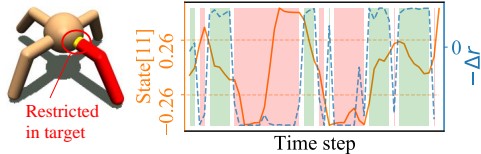

Figure 2: External dynamics shift: *(left)* source and target MDPs (target contains an obstacle represented with the dashed line); *(middle)* top plots (*w/o Aug.*) depict the trajectories that are generated by the learned policy with vanilla MPO; *(middle)* bottom plots (*DARA*) depict the trajectories that are generated by the learned policy with DARA-based MPO; *(right)* learned Q-values on the state-action pairs in left subfigure.

Figure 3: Internal dynamics shift: *(left)* source and target MDPs (range of the right-back-leg of the ant (state[11]) is limited: $[-0.52, 0.52]$ in source MDP $\to [-0.26, 0.26]$ in target MDP); *(right)* the solid (orange) line denotes the state of the right-back-leg over one trajectory collected in source, dashed (blue) line denotes the learned reward modification $-\Delta r$ over the trajectory, and green and red slices denote transition pairs where $-\Delta r \geq$ and $-\Delta r < 0$, *resp.*

Here we characterize both external and internal dynamics shifts: In Map tasks (Figure 2 left), the source dataset $\mathcal{D}'$ is collected in a 2D map and the target $\mathcal{D}$ is collected in the same environment but with an obstacle (the dashed line); In Ant tasks (Figure 3 left), the source dataset $\mathcal{D}'$ is collected using the Mujoco Ant and the target $\mathcal{D}$ is collected with the same Ant but one joint of which is restricted.

Using MPO, as an example of offline RL method, we train a policy on dataset $\{\mathcal{D}' \cup \mathcal{D}\}$ and deploy the acquired policy in both source and target MDPs. As shown in Figure 2 (middle-top, *w/o Aug.*), such training paradigm does not produce an adaptive policy for the target. By modifying rewards in

Table 1: Normalized scores for the (target) D4RL tasks, where our results are averaged over 5 seeds. The arrows in each four-tuple indicate whether the current performance has improved (↑) or not (↓) compared to the previous value. If *1T+10S DARA* achieves comparable (less than 10% degradation) or better performance compared to baseline *10T*, we highlight our scores in bold (in each four-tuple).

| Body Mass Shift | | 10T | 1T | 1T+10S w/o Aug. | 1T+10S DARA | 10T | 1T | 1T+10S w/o Aug. | 1T+10S DARA | 10T | 1T | 1T+10S w/o Aug. | 1T+10S DARA |
|---|---|---|---|---|---|---|---|---|---|---|---|---|---|
| | | BEAR | | | | BRAC-p | | | | AWR | | | |
| Hopper | Random | 11.4 | 1.0↓ | 4.6↑ | 8.4↑ | 11.0 | 10.9↓ | 9.6↓ | **11.0**↑ | 10.2 | 10.3↑ | 3.4↓ | 4.5↑ |
| | Medium | 52.1 | 0.8↓ | 0.9↑ | 1.6↑ | 32.7 | 29.0↓ | 29.2↑ | **32.9**↑ | 35.9 | 30.9↓ | 20.8↓ | 28.9↑ |
| | Medium-R | 33.7 | 1.3↓ | 18.2↑ | **34.1**↑ | 0.6 | 5.4↑ | 20.1↑ | **30.8**↑ | 28.4 | 8.8↓ | 4.1↓ | 4.2↑ |
| | Medium-E | 96.3 | 0.8↓ | 0.6↓ | 1.2↑ | 1.9 | 34.5↑ | 32.3↓ | **34.7**↑ | 27.1 | 27.0↓ | 26.8↓ | **26.6**↓ |
| | | BCQ | | | | CQL | | | | MOPO | | | |
| Hopper | Random | 10.6 | 10.6↓ | 8.3↓ | **9.7**↑ | 10.8 | 10.6↓ | 10.2↓ | **10.4**↑ | 11.7 | 4.8↓ | 2.0↓ | 2.1↑ |
| | Medium | 54.5 | 37.1↓ | 25.7↓ | 38.4↑ | 58.0 | 43.0↓ | 44.9↑ | **59.3**↑ | 28.0 | 4.1↓ | 5.0↑ | 10.7↑ |
| | Medium-R | 33.1 | 9.3↓ | 28.7↑ | **32.8**↑ | 48.6 | 9.6↓ | 1.4↓ | 3.7↑ | 67.5 | 1.0↓ | 5.5↑ | 8.4↑ |
| | Medium-E | 110.9 | 58↓ | 75.4↑ | 84.2↑ | 98.7 | 59.7↓ | 53.6↓ | **99.7**↑ | 23.7 | 1.6↓ | 4.8↑ | 5.8↑ |
| | | BEAR | | | | BRAC-p | | | | AWR | | | |
| Walker2d | Random | 7.3 | 1.5↓ | 3.1↑ | 3.2↑ | -0.2 | 0.0↑ | 1.3↑ | **3.2**↑ | 1.5 | 1.3↓ | 2.0↑ | **2.4**↑ |
| | Medium | 59.1 | -0.5↓ | 0.6↑ | 0.3↓ | 77.5 | 6.4↓ | 70.0↑ | **78.0**↑ | 17.4 | 14.8↓ | 17.1↑ | **17.2**↑ |
| | Medium-R | 19.2 | 0.7↓ | 6.5↑ | 7.3↑ | -0.3 | 8.5↑ | 9.9↑ | **18.6**↑ | 15.5 | 7.4↑ | 1.6↓ | 1.5↓ |
| | Medium-E | 40.1 | -0.1↓ | 1.5↑ | 2.3↑ | 76.9 | 20.6↓ | 64.1↑ | **77.5**↑ | 53.8 | 35.5↓ | 52.5↑ | 53.3↑ |
| | | BCQ | | | | CQL | | | | MOPO | | | |
| Walker2d | Random | 4.9 | 1.8↓ | 4.5↑ | **4.8**↑ | 7.0 | 1.7↓ | 3.2↑ | 3.4↑ | 13.6 | -0.2↓ | -0.1↑ | -0.1↓ |
| | Medium | 53.1 | 32.8↓ | 50.9↑ | **52.3**↑ | 79.2 | 42.9↓ | 80.0↑ | **81.7**↑ | 17.8 | 7.0↓ | 5.7↓ | 11.0↑ |
| | Medium-R | 15.0 | 6.9↓ | 14.9↑ | **15.1**↑ | 26.7 | 4.6↓ | 0.8↓ | 2.0↑ | 39.0 | 5.1↓ | 3.1↓ | 14.2↑ |
| | Medium-E | 57.5 | 32.5↓ | 55.2↑ | **57.2**↑ | 111.0 | 49.5↓ | 63.5↑ | 93.3↑ | 44.6 | 5.3↓ | 5.5↑ | 17.2↑ |

source $\mathcal{D}'$, we show that applying the same training paradigm on the reward augmented data exhibits a positive transfer ability in Figure 2 (middle-bottom, *DARA*). In Figure 2 (right), we show that our DARA produces low Q-values on the obstructive state-action pairs (in left) compared to the vanilla MPO, which thus prevents the Q-value weighted-regression on these unproductive state-action pairs.

More generally, we illustrate how DARA can handle the dynamics adaptation from the reward modification view. In Figure 3 (right), the learned reward modification $-\Delta r$ (dashed blue line) clearly produces a penalty (red slices) on these state-action pairs (in source) that produce infeasible next-state transitions in the target MDP. If we directly apply prior offline RL methods, these transitions that are beyond reach in target and yet are high valued, would yield a negative transfer. Thus, we can think of DARA as finding out these transitions that exhibit dynamics shifts and enabling dynamics adaptation with reward modifications, *e.g.*, penalizing transitions covered by red slices ($-\Delta r < 0$).

## 6.2 CAN DARA ENABLE AN ADAPTIVE POLICY WITH REDUCED OFFLINE DATA IN TARGET?

To characterize the offline dynamics shift, we consider the Hopper, Walker2d and Halfcheetah from the Gym-MuJoCo environment, using offline samples from D4RL as our target offline dataset. For the source dataset, we change the body mass of agents or add joint noise to the motion, and, similar to D4RL, collect the Random, Medium, Medium-R and Medium-E offline datasets for the three environments. Based on various offline RL algorithms (BEAR, BRAC-p, BCQ, CQL, AWR, MOPO), we perform the following comparisons: 1) employing the $100\%$ of D4RL data (*10T*), 2) employing only $10\%$ of the D4RL data (*1T*), 3) employing $10\%$ of the D4RL data and $100\%$ of our collected source offline data (*1T+10S w/o Aug.*), and 4) employing $10\%$ of the D4RL data and $100\%$ of our collected source offline data along with our reward augmentation (*1T+10S DARA*). Due to page limit, here we focus on the dynamics shift concerning the body mass on Walker2d and Hopper. We refer the reader to appendix for more experimental details, tasks, and more baselines (BC, COMBO).

As shown in Table 1, in most of the tasks, the performance degrades substantially when we decrease the amount of target offline data, *i.e.*, *10T → 1T*. Training with additional ten times source offline data (*1T+10S w/o Aug.*) also does not bring substantial improvement (compensating for the reduced data in target), which even degrades the performance in some tasks. We believe that such degradation (compared to *10T*) is caused by the lack of target offline data as well as the dynamics shift (induced by the source data). Incorporating our reward augmentation, we observe that compared to *1T* and *1T+10S w/o Aug.* that both use $10\%$ of the target offline data, our *1T+10S DARA* significantly improves the performance across a majority of tasks. Moreover, DARA can achieve comparable or better performance compared to baseline *10T* that training with ten times as much target offline data.

Table 2: Normalized scores in (target) D4RL tasks, where "Tune" denotes baseline "fine-tune". We observe that with same amount ($10\%$) of target offline data, DARA greatly outperforms baselines.

| Body Mass Shift | | Tune | DARA | Tune | DARA | Tune | DARA | Tune | DARA | Tune | DARA | $\pi_p\hat{T}$ | $\hat{T}\pi_p$ |
|---|---|---|---|---|---|---|---|---|---|---|---|---|---|
| | | BEAR | | BRAC-p | | BCQ | | CQL | | MOPO | | MABE | |
| Hopper | Random | 0.8 | 8.4 ↑ | 6.0 | 11.0 ↑ | 8.8 | 9.7 ↑ | **31.6** | 10.4 ↓ | 0.7 | 2.1 ↑ | 10.6 | 9.0 |
| | Medium | 0.8 | 1.6 ↑ | 22.7 | 32.9 ↑ | 31.7 | 38.4 ↑ | 44.5 | **59.3** ↑ | 0.7 | 10.7 ↑ | 48.8 | 23.1 |
| | Medium-R | 0.7 | **34.1** ↑ | 14.7 | 30.8 ↑ | 27.5 | 32.8 ↑ | 1.3 | 3.7 ↑ | 0.6 | 8.4 ↑ | 17.1 | 20.4 |
| | Medium-E | 0.9 | 1.2 ↑ | 19.2 | 34.7 ↑ | 85.9 | 84.2 ↓ | 47.6 | **99.7** ↑ | 2.2 | 5.8 ↑ | 28.1 | 38.9 |
| | | BEAR | | BRAC-p | | BCQ | | CQL | | MOPO | | MABE | |
| Walker2d | Random | **6.6** | 3.2 ↓ | 3.9 | 3.2 ↓ | 4.7 | 4.8 ↑ | 1.1 | 3.4 ↑ | 0.1 | -0.1 ↓ | 6.0 | -0.2 |
| | Medium | 0.3 | 0.3 ↓ | 76.0 | 78.0 ↑ | 28.4 | 52.3 ↑ | 72.3 | **81.7** ↑ | -0.2 | 11.0 ↑ | 30.1 | 56.7 |
| | Medium-R | 1.2 | 7.3 ↑ | 10.0 | **18.6** ↑ | 10.4 | 15.1 ↑ | 1.8 | 2.0 ↑ | 0.0 | 14.2 ↑ | 13.3 | 12.5 |
| | Medium-E | 2.4 | 2.3 ↓ | 74.5 | 77.5 ↑ | 22.7 | 57.2 ↑ | 68.6 | **93.3** ↑ | 7.3 | 17.2 ↑ | 43.7 | 82.7 |

### 6.3 CAN DARA PERFORM BETTER THAN CROSS-DOMAIN BASELINES?

In Section 6.2, *1T+10S w/o Aug.* does not explicitly learn policy for the target dynamics, thus one proposal (*1T+10S fine-tune*) for adapting the target dynamics is fine-tuning the model that learned with source offline data, using the (reduced) target offline data. Moreover, we also compare DARA with the recently proposed MABE (Cang et al., 2021), which is suitable well for our cross-dynamics setting by introducing behavioral priors $\pi_p$ in the model-based offline setting. Thus, we implement two baselines, 1) *1T+10S MABE $\pi_p\hat{T}$* and 2) *1T+10S MABE $\hat{T}\pi_p$*, which denote 1) learning $\pi_p$ with target domain data and $\hat{T}$ with source domain data, and 2) learning $\pi_p$ with source domain data and $\hat{T}$ with target domain data, respectively. We show the results for the Walker (with body mass shift) in Table 2, and more experiments in Appendix A.3.5. Our results show that DARA achieves significantly better performance than the naïve fine-tune-based approaches in a majority of tasks (67 "↑" vs. 13 "↓", including results in appendix). On twelve out of the sixteen tasks (including results in appendix), DARA-based methods outperform the MABE-based methods. We attribute MABE's failure to the difficulty of the reduced target offline data, which limits the generalization of the learned $\pi_p$ or $\hat{T}$ under such data. However, such reduced data ($10\%$ of target) is sufficient to modify rewards in the source offline data, which thus encourages better performance for our DARA.

***For real-world tasks***, we also test DARA in a new offline dataset on the quadruped robot (see appendix for details). Note that we can not access the privileged information (*e.g.*, coordinate) in real robot, thus the target offline data (collected in real-world) does not contain rewards. This means that prior fine-tune-based and MABE-based methods become unavailable. However, our reward augmentation frees us from the requisite of rewards in

Table 3: Average distance covered in an episode in real robot.

| (BCQ) | w/o Aug. | DARA |
|---|---|---|
| Medium | 0.85 | 1.35 ↑ |
| Medium-E | 1.15 | 1.41 ↑ |
| Medium–R-E | 1.27 | 1.55 ↑ |

target domain. We can freely perform offline training only using the augmented source offline data as long as the learned $\Delta r$ is sufficient. For comparison, we also employ a baseline (*w/o Aug.*): directly deploying the learned policy with source data into the (target) real-world. We present the results (deployed in real with obstructive stairs) in Table 3 and videos in supplementary material. We can observe that training with our reward augmentation, the performance can be substantially improved. Due to page limit, we refer readers to Appendix A.3.6 for more experimental results and discussion.

## 7 CONCLUSION

In this paper, we formulate the dynamics shift in offline RL. Based on prior model-based and model-free offline algorithms, we propose the dynamics-aware reward augmentation (DARA) framework that characterizes constraints over state-action-next-state distributions. Empirically we demonstrate DARA can eliminate the dynamics shift and outperform baselines in simulated and real-world tasks.

In Appendix A.2, we characterize our dynamics-aware reward augmentation from the density regularization view, which shows that it is straightforward to derive the reward modification built on prior regularized max-return objective *e.g.*, AlgaeDICE (Nachum et al., 2019b). We list some related works in Table 4, where the majority of the existing work focuses on regularizing state-action distribution, while dynamics shift receives relatively little attention. Thus, we hope to shift the focus of the community towards analyzing how dynamics shift affects RL and how to eliminate the effect.

REPRODUCIBILITY STATEMENT

Our experimental evaluation is conducted with publicly available D4RL (Fu et al., 2020) and Ne-oRL (Qin et al., 2021). In Appendix A.4 and A.5, we provide the environmental details and training setup for our real-world sim2real tasks. In supplementary material, we upload our source code and the collected offline dataset for the the quadruped robot.

ACKNOWLEDGMENTS

We thank Zifeng Zhuang, Yachen Kang and Qiangxing Tian for helpful feedback and discussions. This work is supported by NSFC General Program (62176215).

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

## A APPENDIX

### A.1 DERIVATION

#### A.1.1 PROOF OF LEMMA 3

Let $\mathcal{B}_M^\pi V(\mathbf{s}) = \mathbb{E}_{\mathbf{a} \sim \pi(\mathbf{a}|\mathbf{s})} \left[ r(\mathbf{s}, \mathbf{a}) + \gamma \mathbb{E}_{\mathbf{s}' \sim T(\mathbf{s}'|\mathbf{s}, \mathbf{a})} \left[ V(\mathbf{s}') \right] \right]$ and $r(\mathbf{s}) = \mathbb{E}_{\mathbf{a} \sim \pi(\mathbf{a}|\mathbf{s})} \left[ r(\mathbf{s}, \mathbf{a}) \right]$. Then, we have

$$
\begin{aligned}
\eta_{\hat{M}'}(\pi) - \eta_M(\pi) &= \mathbb{E}_{\mathbf{s}_0 \sim \rho_0(\mathbf{s})} \left[ V_{\hat{M}'}(\mathbf{s}_0) - V_M(\mathbf{s}_0) \right] \\
&= \sum_{t=0}^\infty \gamma^t \mathbb{E}_{\mathbf{s}_t \sim P(\mathbf{s}_t|\pi, \hat{M}', t)} \mathbb{E}_{\mathbf{a}_t \sim \pi(\mathbf{a}_t|\mathbf{s}_t)} \left[ r(\mathbf{s}_t, \mathbf{a}_t) \right] - \mathbb{E}_{\mathbf{s}_0 \sim \rho_0(\mathbf{s})} \left[ V_M(\mathbf{s}_0) \right] \\
&= \sum_{t=0}^\infty \gamma^t \mathbb{E}_{\mathbf{s}_t \sim P(\mathbf{s}_t|\pi, \hat{M}', t)} \left[ r(\mathbf{s}_t) + V_M(\mathbf{s}_t) - V_M(\mathbf{s}_t) \right] - \mathbb{E}_{\mathbf{s}_0 \sim \rho_0(\mathbf{s})} \left[ V_M(\mathbf{s}_0) \right] \\
&= \sum_{t=0}^\infty \gamma^t \mathbb{E}_{\substack{\mathbf{s}_t \sim P(\mathbf{s}_t|\pi, \hat{M}', t) \\ \mathbf{s}_{t+1} \sim P(\mathbf{s}_{t+1}|\pi, \hat{M}', t+1)}} \left[ r(\mathbf{s}_t) + \gamma V_M(\mathbf{s}_{t+1}) - V_M(\mathbf{s}_t) \right] \\
&= \sum_{t=0}^\infty \gamma^t \mathbb{E}_{\substack{\mathbf{s}_t \sim P(\mathbf{s}_t|\pi, \hat{M}', t) \\ \mathbf{s}_{t+1} \sim P(\mathbf{s}_{t+1}|\pi, \hat{M}', t+1)}} \left[ r(\mathbf{s}_t) + \gamma V_M(\mathbf{s}_{t+1}) - \left( r(\mathbf{s}_t) + \gamma \mathbb{E}_{\mathbf{a} \sim \pi(\mathbf{a}|\mathbf{s}_t), \mathbf{s}' \sim T(\mathbf{s}_t, \mathbf{a})} \left[ V_M(\mathbf{s}') \right] \right) \right] \\
&= \sum_{t=0}^\infty \gamma^t \mathbb{E}_{\mathbf{s}_t \sim P(\mathbf{s}_t|\pi, \hat{M}', t)} \left[ \mathcal{B}_{\hat{M}'}^\pi V_M(\mathbf{s}_t) - \mathcal{B}_M^\pi V_M(\mathbf{s}_t) \right] \\
&= \mathbb{E}_{\mathbf{s} \sim d_{\hat{M}'}^\pi(\mathbf{s})} \left[ \mathcal{B}_{\hat{M}'}^\pi V_M(\mathbf{s}) - \mathcal{B}_M^\pi V_M(\mathbf{s}) \right].
\end{aligned}
$$

#### A.1.2 MODEL-BASED FORMULATION

Here we provide detailed derivation of the lower bound in Equation 9 in the main text.

**Assumption 1** *Assume a scale $c$ and a function class $\mathcal{F}$ such that $V_M \in c\mathcal{F}$.*

Following MOPO (Yu et al., 2020), we set $\mathcal{F} = \{f : \|f\|_\infty \le 1\}$. In Section Preliminaries, we have that the reward function is bounded: $r(\mathbf{s}, \mathbf{a}) \in [-R_{max}, R_{max}]$. Thus, we have $\|V_M\|_\infty \le \sum_{t=0}^\infty \gamma^t R_{max} = \frac{R_{max}}{1-\gamma}$ and hence the scale $c = \frac{R_{max}}{1-\gamma}$.

As a direct corollary of Assumption 1 and Equation 5, we have

$$
\left| \mathbb{E}_{\mathbf{s} \sim d_{\hat{M}'}^\pi(\mathbf{s})} \left[ \mathcal{B}_{\hat{M}'}^\pi V_M(\mathbf{s}) - \mathcal{B}_M^\pi V_M(\mathbf{s}) \right] \right| \le \gamma c \cdot \mathbb{E}_{\mathbf{s}, \mathbf{a} \sim d_{\hat{M}'}^\pi(\mathbf{s}) \pi(\mathbf{a}|\mathbf{s})} \left[ d_{\mathcal{F}}(\hat{T}'(\mathbf{s}'|\mathbf{s}, \mathbf{a}), T(\mathbf{s}'|\mathbf{s}, \mathbf{a})) \right]. \tag{10}
$$

Further, we find

$$
d_{\mathcal{F}}(\hat{T}'(\mathbf{s}'|\mathbf{s}, \mathbf{a}), T(\mathbf{s}'|\mathbf{s}, \mathbf{a})) \le d_{\mathcal{F}}(\hat{T}'(\mathbf{s}'|\mathbf{s}, \mathbf{a}), \hat{T}(\mathbf{s}'|\mathbf{s}, \mathbf{a})) + d_{\mathcal{F}}(\hat{T}(\mathbf{s}'|\mathbf{s}, \mathbf{a}), T(\mathbf{s}'|\mathbf{s}, \mathbf{a})) \tag{11}
$$

For the first term $d_{\mathcal{F}}(\hat{T}'(\mathbf{s}'|\mathbf{s}, \mathbf{a}), \hat{T}(\mathbf{s}'|\mathbf{s}, \mathbf{a}))$ in Equation 12, through Pinsker's inequality, we have

$$
d_{\mathcal{F}}(\hat{T}'(\mathbf{s}'|\mathbf{s}, \mathbf{a}), \hat{T}(\mathbf{s}'|\mathbf{s}, \mathbf{a})) = D_{\text{TV}}(\hat{T}'(\mathbf{s}'|\mathbf{s}, \mathbf{a}), \hat{T}(\mathbf{s}'|\mathbf{s}, \mathbf{a})) \le \sqrt{\frac{1}{2} D_{\text{KL}}(\hat{T}'(\mathbf{s}'|\mathbf{s}, \mathbf{a}), \hat{T}(\mathbf{s}'|\mathbf{s}, \mathbf{a}))} \tag{12}
$$

To keep consistent with the DARA-based method-free offline methods, we introduce scale $\eta$ and bias $\delta$ to eliminate the square root in Equation 12. To be specific, we assume[2] scale $\eta$ and bias $\delta$ such that $\sqrt{\frac{1}{2} D_{\text{KL}}(\hat{T}', \hat{T})} \le \eta D_{\text{KL}}(\hat{T}', \hat{T}) + \delta$. Thus, we obtain

$$
d_{\mathcal{F}}(\hat{T}'(\mathbf{s}'|\mathbf{s}, \mathbf{a}), \hat{T}(\mathbf{s}'|\mathbf{s}, \mathbf{a})) = D_{\text{TV}}(\hat{T}'(\mathbf{s}'|\mathbf{s}, \mathbf{a}), \hat{T}(\mathbf{s}'|\mathbf{s}, \mathbf{a})) \le \eta D_{\text{KL}}(\hat{T}'(\mathbf{s}'|\mathbf{s}, \mathbf{a}), \hat{T}(\mathbf{s}'|\mathbf{s}, \mathbf{a})) + \delta \tag{13}
$$

---

[2]In implementation, we clip the maximum deviation of $\log \frac{\hat{T}'(\mathbf{s}'|\mathbf{s}, \mathbf{a})}{\hat{T}(\mathbf{s}'|\mathbf{s}, \mathbf{a})}$ for each $(\mathbf{s}, \mathbf{a}, \mathbf{s}')$, which thus makes $D_{\text{KL}}(\hat{T}'(\mathbf{s}'|\mathbf{s}, \mathbf{a}), \hat{T}(\mathbf{s}'|\mathbf{s}, \mathbf{a}))$ bounded.

For the second term $d_{\mathcal{F}}(\hat{T}(\mathbf{s}'|\mathbf{s}, \mathbf{a}), T(\mathbf{s}'|\mathbf{s}, \mathbf{a}))$ in Equation 11, we assume that we have access to an oracle uncertainty qualification module that provides an upper bound on the error of the estimated empirical MDP $\hat{M} := \{\mathcal{S}, \mathcal{A}, r, \hat{T}, \rho_0, \gamma\}$.

**Assumption 2** *Let $\mathcal{F}$ be the function class in Assumption 1. We say $u : \mathcal{S} \times \mathcal{A} \to \mathbb{R}$ is an admissible error estimator for $\hat{T}$ if $d_{\mathcal{F}}(\hat{T}(\mathbf{s}'|\mathbf{s}, \mathbf{a}), T(\mathbf{s}'|\mathbf{s}, \mathbf{a})) \leq u(\mathbf{s}, \mathbf{a})$ for all $(\mathbf{s}, \mathbf{a})$.*

Thus, we have

$$\mathbb{E}_{\mathbf{s}, \mathbf{a} \sim d_{\hat{M}'}^{\pi}(\mathbf{s})\pi(\mathbf{a}|\mathbf{s})} \left[ d_{\mathcal{F}}(\hat{T}(\mathbf{s}'|\mathbf{s}, \mathbf{a}), T(\mathbf{s}'|\mathbf{s}, \mathbf{a})) \right] \leq \mathbb{E}_{\mathbf{s}, \mathbf{a} \sim d_{\hat{M}'}^{\pi}(\mathbf{s})\pi(\mathbf{a}|\mathbf{s})} \left[ u(\mathbf{s}, \mathbf{a}) \right] \tag{14}$$

Bring Inequations 10, 11, 13, and 14 into Lemma 3, we thus have

$$\eta_M(\pi) \geq \mathbb{E}_{\mathbf{s}, \mathbf{a}, \mathbf{s}' \sim d_{\hat{M}'}^{\pi}(\mathbf{s})\pi(\mathbf{a}|\mathbf{s})\hat{T}'(\mathbf{s}'|\mathbf{s}, \mathbf{a})} \left[ r(\mathbf{s}, \mathbf{a}) - \eta\gamma c \log \frac{\hat{T}'(\mathbf{s}'|\mathbf{s}, \mathbf{a})}{\hat{T}(\mathbf{s}'|\mathbf{s}, \mathbf{a})} - \gamma c u(\mathbf{s}, \mathbf{a}) - \gamma c \delta \right]. \tag{15}$$

### A.1.3 LEARNING CLASSIFIERS

Applying Bayes' rule, we have

$$\hat{T}'(\mathbf{s}'|\mathbf{a}, \mathbf{s}) := p(\mathbf{s}'|\mathbf{s}, \mathbf{a}, \text{source}) = \frac{p(\text{source}|\mathbf{s}, \mathbf{a}, \mathbf{s}')p(\mathbf{s}, \mathbf{a}, \mathbf{s}')}{p(\text{source}|\mathbf{s}, \mathbf{a})p(\mathbf{s}, \mathbf{a})},$$

$$\hat{T}(\mathbf{s}'|\mathbf{a}, \mathbf{s}) := p(\mathbf{s}'|\mathbf{s}, \mathbf{a}, \text{target}) = \frac{p(\text{target}|\mathbf{s}, \mathbf{a}, \mathbf{s}')p(\mathbf{s}, \mathbf{a}, \mathbf{s}')}{p(\text{target}|\mathbf{s}, \mathbf{a})p(\mathbf{s}, \mathbf{a})}.$$

Then we parameterize $p(\cdot|\mathbf{s}, \mathbf{a}, \mathbf{s}')$ and $p(\cdot|\mathbf{s}, \mathbf{a})$ with the two classifiers $q_{\text{sas}}$ and $q_{\text{sa}}$ respectively. Using the standard cross-entropy loss, we learn $q_{\text{sas}}$ and $q_{\text{sa}}$ with the following optimization objective:

$$\max \quad \mathbb{E}_{(\mathbf{s}, \mathbf{a}, \mathbf{s}') \sim \mathcal{D}'} \left[ \log q_{\text{sas}}(\text{source}|\mathbf{s}, \mathbf{a}, \mathbf{s}') \right] + \mathbb{E}_{(\mathbf{s}, \mathbf{a}, \mathbf{s}') \sim \mathcal{D}} \left[ \log q_{\text{sas}}(\text{target}|\mathbf{s}, \mathbf{a}, \mathbf{s}') \right],$$

$$\max \quad \mathbb{E}_{(\mathbf{s}, \mathbf{a}) \sim \mathcal{D}'} \left[ \log q_{\text{sa}}(\text{source}|\mathbf{s}, \mathbf{a}) \right] + \mathbb{E}_{(\mathbf{s}, \mathbf{a}) \sim \mathcal{D}} \left[ \log q_{\text{sa}}(\text{target}|\mathbf{s}, \mathbf{a}) \right].$$

With the trained $q_{\text{sas}}$ and $q_{\text{sa}}$, we have

$$\log \frac{\hat{T}'(\mathbf{s}'|\mathbf{s}, \mathbf{a})}{\hat{T}(\mathbf{s}'|\mathbf{s}, \mathbf{a})} = \log \frac{q_{\text{sas}}(\text{source}|\mathbf{s}, \mathbf{a}, \mathbf{s}')}{q_{\text{sas}}(\text{target}|\mathbf{s}, \mathbf{a}, \mathbf{s}')} - \log \frac{q_{\text{sa}}(\text{source}|\mathbf{s}, \mathbf{a})}{q_{\text{sa}}(\text{target}|\mathbf{s}, \mathbf{a})}. \tag{16}$$

In our implementation, we also clip the above reward modification between $-10$ and $10$.

### A.2 REGULARIZATION VIEW OF DYNAMICS-AWARE REWARD AUGMENTATION

Here we shortly characterize our dynamics-aware reward augmentation from the density regularization. Note the standard max-return objective $\eta_M(\pi)$ in RL can be written exclusively in terms of the on-policy distribution $d_M^{\pi}(\mathbf{s})\pi(\mathbf{a}|\mathbf{s})$. To introduce an off-policy distribution $d_{\mathcal{D}}(\mathbf{s})\pi_b(\mathbf{a}|\mathbf{s})$ in the objective, prior works often incorporate a regularization (penalty): $D(d_M^{\pi}(\mathbf{s})\pi(\mathbf{a}|\mathbf{s})\|d_{\mathcal{D}}(\mathbf{s})\pi_b(\mathbf{a}|\mathbf{s}))$, as in Equations 2 and 3. However, facing dynamics shift, such regularization should take into account the transition dynamics, which is penalizing $D(d_M^{\pi}(\mathbf{s})\pi(\mathbf{a}|\mathbf{s})T(\mathbf{s}'|\mathbf{s}, \mathbf{a})\|d_{\mathcal{D}'}(\mathbf{s})\pi_{b'}(\mathbf{a}|\mathbf{s})\hat{T}'(\mathbf{s}'|\mathbf{s}, \mathbf{a}))$. From this view, it is also straightforward to derive the reward modification built on prior regularized off-policy max-return objective *e.g.*, the off-policy approach AlgaeDICE (Nachum et al., 2019b).

In Table 4, we provide some related works with respect to the (state-action pair) $d_{\mathcal{D}}(\mathbf{s})\pi_b(\mathbf{a}|\mathbf{s})$ regularization and the (state-action-next-state pair) $d_{\mathcal{D}'}(\mathbf{s})\pi_{b'}(\mathbf{a}|\mathbf{s})\hat{T}'(\mathbf{s}'|\mathbf{s}, \mathbf{a})$ regularization. We can find that the majority of the existing work focuses on regularizing state-action distribution, while dynamics shift receives relatively little attention. Thus, we hope to shift the focus of the community towards analyzing how the dynamics shift affects RL and how to eliminate the effect.

Table 4: Some related works with **explicit** (state-action $p(\mathbf{s}, \mathbf{a})$ or state-action-next-state $p(\mathbf{s}, \mathbf{a}, \mathbf{s}')$) **regularization**. More papers with respect to unsupervised RL, inverse RL (imitation learning), meta RL, multi-agent RL, and hierarchical RL are not included.

| reg. with $d_{\mathcal{D}}(\mathbf{s})\pi_b(\mathbf{a}|\mathbf{s})$ | | reg. with $d_{\mathcal{D}'}(\mathbf{s})\pi_{b'}(\mathbf{a}|\mathbf{s})\hat{T}'(\mathbf{s}'|\mathbf{s}, \mathbf{a})$ |
|---|---|---|
| *Online:* 
 see summarization in Geist et al. (2019) and Vieillard et al. (2020). | | Eysenbach et al. (2021) (DARC) 
 Liu et al. (2021) (DARS); 
 Haghgoo et al. (2021) |
| *Offline (off-policy evaluation):* | | |
| Fujimoto et al. (2019) (BCQ); | Kumar et al. (2019) (BEAR); | |
| Wu et al. (2019) (BRAC-p); | Abdolmaleki et al. (2018) (MPO); | |
| Peng et al. (2019) (AWR); | Nair et al. (2020) (AWAC); | |
| Wang et al. (2020) (CRR); | Siegel et al. (2020); | |
| Chen et al. (2019); (BAIL) | Kumar et al. (2020a) (CQL); | |
| Xu et al. (2021) (CPQ); | Kostrikov et al. (2021) (Fisher-BRC); | |
| Liu et al. (2018); | Nachum et al. (2019a) (DualDICE); | |
| Nachum et al. (2019b) (AlgaeDICE); | Zhang et al. (2020) (GenDICE); | |
| Yang et al. (2020); | Nachum & Dai (2020); | |
| Jiang & Huang (2020); | Uehara et al. (2020); | |
| Yu et al. (2020) (MOPO); | Kidambi et al. (2020) (MOReL); | |
| Yu et al. (2021) (COMBO); | Cang et al. (2021) (MABE); | |

## A.3 MORE EXPERIMENTS

### A.3.1 TRAINING WITH $\{\mathcal{D}' \cup \mathcal{D}\}$

As we show in Figure 1 in Section Introduction, the performance of prior offline RL methods deteriorates dramatically as the amount of (target) offline data $\mathcal{D}$ decreases. In Figure 4, we show that directly training with the mixed dataset $\{\mathcal{D}' \cup \mathcal{D}\}$ will not compensate for the deteriorated performance caused by the reduced target offline data, and training with such additional source offline data can even lead the performance degradation in some tasks.

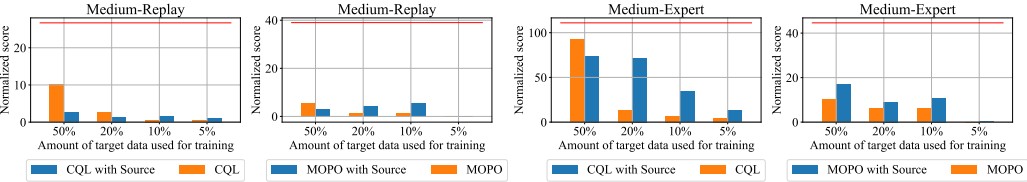

Figure 4: Final performance on the D4RL (Walker2d) task: The orange bars denote the final performance with different amount ($50\%\mathcal{D}$, $20\%\mathcal{D}$, $10\%\mathcal{D}$, $5\%\mathcal{D}$) of target offline data; The blue bars denote the final performance of mixing $100\%$ of source offline data $\mathcal{D}'$ and different amount of target data $x\%\mathcal{D}$ ($x \in [50, 20, 10, 5]$), *i.e.*, training with $\{100\%\mathcal{D}' \cup x\%\mathcal{D}\}$; The red lines denote the final performance of training with $100\%$ of target offline data $\mathcal{D}$. We can observe that 1) the performance deteriorates dramatically as the amount of (target) offline data decreases ($100\%\mathcal{D}$ (red line) $\rightarrow$ $50\%\mathcal{D}$ (orange bar) $\rightarrow 20\%\mathcal{D}$ (orange bar) $\rightarrow 10\%\mathcal{D}$ (orange bar) $\rightarrow 5\%\mathcal{D}$ (orange bar)), 2) after training with the additional $100\%$ of source offline data, $\{100\%\mathcal{D}' \cup x\%\mathcal{D}\}$, the final performance is improved in some tasks, but most of the improvement is a pittance compared to the original performance degradation (compared to that training with the $100\%$ of target offline data, *i.e.*, the red lines), and 3) what is worse is that adding source offline data $\mathcal{D}'$ even leads performance degradation in some tasks, *e.g.*, CQL with $50\%\mathcal{D}$ and $20\%\mathcal{D}$ in Medium-Random.

### A.3.2 COMPARISON BETWEEN LEARNING CLASSIFIERS AND LEARNING DYNAMICS (FOR THE REWARD MODIFICATION)

Table 5: Normalized scores for the Hopper tasks with the body mass (dynamics) shift. Rat. and Cla. denote estimating the reward modification with the estimated-dynamics ratio and learned classifiers (Appendix A.1.3), respectively.

| Body Mass Shift | | BEAR | | BRAC-p | | AWR | | BCQ | | CQL | | MOPO | |
|---|---|---|---|---|---|---|---|---|---|---|---|---|---|
| | | Rat. | Cla. | Cla. | Cla. | Rat. | Cla. | Rat. | Cla. | Rat. | Cla. | Rat. | Cla. |
| Hopper | Random | 9.9 > 8.4 | | 11.2 > | 11.0 | 3.7 < | 4.5 | 8.5 < | 9.7 | 11.8 > | 10.4 | 1.8 < | 2.1 |
| | Medium | 0.8 < 1.6 | | 31.7 < | 32.9 | 18.0 < | 28.9 | 33.2 < | 38.4 | 45.9 < | 59.3 | 3.1 < | 10.7 |
| | Medium-R | 28.4 < 34.1 | | 36.5 > | 30.8 | 2.5 < | 4.2 | 33.9 > | 32.8 | 2.0 < | 3.7 | 3.8 < | 8.4 |
| | Medium-E | 0.8 < 1.2 | | 50.9 > | 34.7 | 45.8 < | 26.6 | 68.4 < | 84.2 | 107.3 > | 99.7 | 5.7 < | 5.8 |

In Table 5, we show the comparison between learning classifiers and learning dynamics (for our reward modification) in the Hopper tasks. We can observe that the two schemes for estimating the reward modification have similar performance. Thus, for simplicity and following Eysenbach et al. (2021), we adopt the classifiers to modify rewards in the source offline data in our experiments.

### A.3.3 MORE EXAMPLES WITH RESPECT TO THE REWARD AUGMENTATION

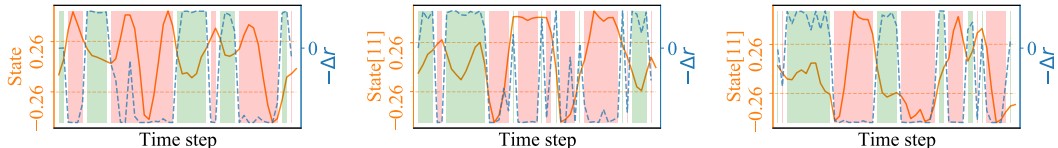

Figure 5: We can observe that our reward augmentation 1) encourages ($-\Delta r > 0$, *i.e.*, the green slice parts) these transitions ($-0.26 \leq$ next-state$[11] \leq 0.26$) that have the same dynamics with the target environment, and 2) discourages ($-\Delta r < 0$, *i.e.*, the red slice parts) these transitions that have different (unreachable) dynamics (next-state$[11] \leq -0.26$ or next-state$[11] \geq 0.26$) in the target.

In Figure 5, we provide more examples with respect to the reward augmentation in the Ant task in Figure 3 (left).

### A.3.4 COMPARISON BETWEEN *10T*, *1T*, *1T+10S w/o Aug.*, AND *1T+10S DARA*

Based on various offline RL algorithms (BEAR (Kumar et al., 2019), BRAC-p (Wu et al., 2019), BCQ (Fujimoto et al., 2019), CQL (Kumar et al., 2020a), AWR (Peng et al., 2019), MOPO (Yu et al., 2020), BC (behavior cloning), COMBO (Yu et al., 2021)), we provide the additional results in Tables 6, 7, 8, 9, and 10.

Table 6: Normalized scores for the Hopper tasks with the body mass (dynamics) shift. (The comparison results for BEAR, BRAC-p, AWR, CQL, and MOPO are provided in the main text.)

| Body Mass Shift | | 10T | 1T | 1T+10S w/o Aug. | 1T+10S DARA | 10T | 1T | 1T+10S w/o Aug. | 1T+10S DARA | 10T | 1T | 1T+10S w/o Aug. | 1T+10S DARA |
|---|---|---|---|---|---|---|---|---|---|---|---|---|---|
| | | | | BC | | | | COMBO | | | | | |
| Hopper | Random | 9.8 | 9.8 ↑ | 6.9 ↓ | 10.1 ↑ | 17.9 | 0.7 ↓ | 5.4 ↑ | 4.6 ↓ | | | | |
| | Medium | 29.0 | 27.9 ↓ | 17.6 ↓ | 25.0 ↑ | 94.9 | 1.8 ↓ | 33.7 ↓ | 45.7 ↑ | | | | |
| | Medium-R | 11.8 | 7.8 ↓ | 7.7 ↓ | 11.6 ↑ | 73.1 | 13.1 ↓ | 11.0 ↓ | 27.9 ↑ | | | | |
| | Medium-E | 111.9 | 21.5 ↓ | 20.8 ↓ | 35.7 ↑ | 111.1 | 0.8 ↓ | 14.9 ↑ | 108.1 ↑ | | | | |

Table 7: Normalized scores for the Hopper tasks with the joint noise (dynamics) shift.

| Joint Noise Shift | 10T | 1T | 1T+10S w/o Aug. | 1T+10S DARA | 10T | 1T | 1T+10S w/o Aug. | 1T+10S DARA | 10T | 1T | 1T+10S w/o Aug. | 1T+10S DARA |
|---|---|---|---|---|---|---|---|---|---|---|---|---|
| | BEAR | | | | BRAC-p | | | | AWR | | | |
| **Hopper** Random | 11.4 | 0.6↓ | 7.4↑ | 4.2↓ | 11.0 | 10.8↓ | 10.0↓ | 10.8↑ | 10.2 | 10.1↓ | 3.6↓ | 4.0↑ |
| Medium | 52.1 | 0.8↓ | 2.0↑ | 2.0↓ | 32.7 | 26.6↓ | 27.6↑ | 37.6↑ | 35.9 | 30.3↓ | 38.8↑ | 41.3↑ |
| Medium-R | 33.7 | 2.7↓ | 3.6↑ | 9.9↑ | 0.6 | 13.4↑ | 89.9↑ | 101.4↑ | 28.4 | 12.4↓ | 6.7↑ | 7.2↑ |
| Medium-E | 96.3 | 0.8↓ | 0.8↓ | 1.4↑ | 1.9 | 19.8↑ | 57.6↑ | 87.8↑ | 27.1 | 25.5↓ | 27.0↑ | 27.0↓ |
| | BCQ | | | | CQL | | | | MOPO | | | |
| **Hopper** Random | 10.6 | 10.5↓ | 7.0↓ | 9.6↑ | 10.8 | 10.4↓ | 10.4↓ | 10.8↓ | 11.7 | 1.5↓ | 1.3↓ | 2.9↑ |
| Medium | 54.5 | 45.8↓ | 49.0↑ | 54.4↑ | 58.0 | 46.2↓ | 58.0↑ | 58.0↓ | 28.0 | 2.7↓ | 9.2↑ | 17.3↑ |
| Medium-R | 33.1 | 13.0↓ | 23.8↑ | 32.0↑ | 48.6 | 13.6↓ | 2.6↓ | 3.6↑ | 67.5 | 0.8↓ | 2.3↑ | 6.4↑ |
| Medium-E | 110.9 | 44.6↓ | 96↑ | 109↑ | 98.7 | 50.7↓ | 73.4↑ | 108.9↑ | 23.7 | 1↓ | 6.1↑ | 7.5↑ |
| | BC | | | | COMBO | | | | | | | |
| **Hopper** Random | 9.8 | 9.8↑ | 7.5↓ | 9.1↑ | 17.9 | 0.7↓ | 1.8↑ | 4.9↑ | | | | |
| Medium | 29.0 | 27.9↓ | 29.0↑ | 29.0↑ | 94.9 | 1.8↓ | 0.7↓ | 9.6↑ | | | | |
| Medium-R | 11.8 | 7.8↓ | 8.5↑ | 11.3↑ | 73.1 | 13.1↓ | 4.0↓ | 9.6↑ | | | | |
| Medium-E | 111.9 | 21.5↓ | 53.5↑ | 77.9↑ | 111.1 | 0.8↓ | 34.0↑ | 45.9↑ | | | | |

Table 8: Normalized scores for the Walker2d tasks with the body mass (dynamics) shift. (The comparison results for BEAR, BRAC-p, AWR, CQL, and MOPO are provided in the main text.)

| Body Mass Shift | 10T | 1T | 1T+10S w/o Aug. | 1T+10S DARA | 10T | 1T | 1T+10S w/o Aug. | 1T+10S DARA | 10T | 1T | 1T+10S w/o Aug. | 1T+10S DARA |
|---|---|---|---|---|---|---|---|---|---|---|---|---|
| | BC | | | | COMBO | | | | | | | |
| **Walker2d** Random | 1.6 | 0.1↓ | 1.7↑ | 2.7↑ | 7.0 | 1.8↓ | 2.0↑ | 3.5↑ | | | | |
| Medium | 6.6 | 5.5↓ | 3.8↓ | 6.6↑ | 75.5 | -1.0↓ | 23.9↑ | 36.6↑ | | | | |
| Medium-R | 11.3 | 6.6↓ | 8.1↑ | 11.0↑ | 56.0 | 0.1↓ | 11.4↑ | 22.6↑ | | | | |
| Medium-E | 6.4 | 3.1↓ | 6.0↑ | 6.2↑ | 96.1 | -0.9↓ | -0.1↑ | -0.1↑ | | | | |

Table 9: Normalized scores for the Walker2d tasks with the joint noise (dynamics) shift.

| Joint Noise Shift | 10T | 1T | 1T+10S w/o Aug. | 1T+10S DARA | 10T | 1T | 1T+10S w/o Aug. | 1T+10S DARA | 10T | 1T | 1T+10S w/o Aug. | 1T+10S DARA |
|---|---|---|---|---|---|---|---|---|---|---|---|---|
| | BEAR | | | | BRAC-p | | | | AWR | | | |
| **Walker2d** Random | 7.3 | 2.2↓ | 0.6↓ | 2.6↑ | -0.2 | 2.8↑ | 3.3↑ | 8.8↑ | 1.5 | 0.9↓ | 1.5↑ | 1.5↓ |
| Medium | 59.1 | -0.4↓ | 0.6↑ | 0.1↑ | 77.5 | 28.8↓ | 55.2↑ | 72.9↑ | 17.4 | 12.2↓ | 17.2↑ | 17.2↓ |
| Medium-R | 19.2 | 0.4↓ | 4↑ | 10.4↑ | -0.3 | 6.3↑ | 32.1↑ | 34.8↑ | 15.5 | 6↓ | 1.4↓ | 2.1↑ |
| Medium-E | 40.1 | -0.2↓ | 0.8↑ | 0.6↑ | 76.9 | 21.8↓ | 62.3↑ | 74.3↑ | 53.8 | 40.4↓ | 53↑ | 53.6↑ |
| | BCQ | | | | CQL | | | | MOPO | | | |
| **Walker2d** Random | 4.9 | 3.7↓ | 3.4↓ | 5.2↑ | 7 | 0.5↓ | 2.7↑ | 6.4↑ | 13.6 | -0.3↓ | -0.2↑ | -0.2↓ |
| Medium | 53.1 | 43↓ | 44.9↑ | 52.7↑ | 79.2 | 43.9↓ | 73.2↑ | 81.2↑ | 17.8 | 5.8↓ | 7.8↑ | 12.2↑ |
| Medium-R | 15 | 5.7↓ | 9.8↑ | 14.6↑ | 26.7 | 1.8↓ | 1.4↓ | 1.8↑ | 39 | 0.8↓ | 9.3↑ | 16.4↑ |
| Medium-E | 57.5 | 44.5↓ | 40.6↓ | 57.2↑ | 111 | 46.8↓ | 109.9↑ | 116.5↑ | 44.6 | 2.9↓ | 15.2↑ | 26.3↑ |
| | BC | | | | COMBO | | | | | | | |
| **Walker2d** Random | 1.6 | 0.1↓ | 0.9↑ | 1.6↑ | 7.0 | 1.8↓ | 0.1↓ | 1.5↑ | | | | |
| Medium | 6.6 | 5.5↓ | 6.4↑ | 6.5↑ | 75.5 | -1.0↓ | 0.4↑ | 0.7↑ | | | | |
| Medium-R | 11.3 | 6.6↓ | 4.6↓ | 10.4↑ | 56.0 | 0.1↓ | 5.6↑ | 7.4↑ | | | | |
| Medium-E | 6.4 | 3.1↓ | 6.2↑ | 6.4↑ | 96.1 | -0.9↓ | 0.8↑ | -0.1↓ | | | | |

Table 10: Normalized scores for the Halfcheetah tasks with the joint noise (dynamics) shift.

| | Joint Noise Shift | 10T | 1T | 1T+10S w/o Aug. | 1T+10S DARA | 10T | 1T | 1T+10S w/o Aug. | 1T+10S DARA | 10T | 1T | 1T+10S w/o Aug. | 1T+10S DARA |
|---|---|---|---|---|---|---|---|---|---|---|---|---|---|
| | | BEAR | | | | BRAC-p | | | | AWR | | | |
| Halfcheetah | Random | 25.1 | 17.8↓ | 25.0↑ | 25.1↑ | 24.1 | 10.0↓ | 25.0↑ | 26.7↑ | 2.5 | 2.7↑ | 3.1↑ | 48.9↑ |
| | Medium | 41.7 | -0.2↓ | 0.8↑ | 1.5↑ | 43.8 | 43.0↓ | 52.4↑ | 53.0↑ | 37.4 | 38.2↑ | 48.7↑ | 37.4↓ |
| | Medium-R | 38.6 | 9.3↓ | -0.6↓ | -0.5↑ | 45.4 | 2.5↓ | -2.3↓ | 45.3↑ | 40.3 | 2.6↓ | 2.3↓ | 2.3↓ |
| | Medium-E | 53.4 | -1.2↓ | 1.0↑ | -1.4↓ | 44.2 | 6.9↓ | 0.9↓ | 45.3↑ | 52.7 | 32.2↓ | 80.6↑ | 79.2↓ |
| | | BCQ | | | | CQL | | | | MOPO | | | |
| Halfcheetah | Random | 2.2 | 2.3↑ | 2.2↓ | 2.3↑ | 35.4 | -2.3↓ | -2.4↓ | 10.4↑ | 35.4 | 2.3↓ | 1.2↓ | 1.1↓ |
| | Medium | 40.7 | 37.6↓ | 40.0↑ | 48.6↑ | 44.4 | 35.4↓ | 40.7↑ | 52.6↑ | 42.3 | 3.2↓ | 3.5↓ | 5.3↑ |
| | Medium-R | 38.2 | 1.1↓ | 39.4↑ | 41.3↑ | 46.2 | 0.6↓ | 2.0↑ | 1.9↓ | 53.1 | -0.1↓ | 2.6↑ | 4.2↑ |
| | Medium-E | 64.7 | 37.3↓ | 55.3↑ | 76.9↑ | 62.4 | -3.3↓ | 7.7↑ | 1.7↓ | 63.3 | 4.2↓ | 1.5↓ | 7.2↑ |
| | | BC | | | | COMBO | | | | | | | |
| Halfcheetah | Random | 2.1 | 2.0↓ | 2.2↑ | 2.2↑ | 38.8 | 24.0↓ | 18.7↓ | 20.3↑ | | | | |
| | Medium | 36.1 | 36.5↑ | 49.4↑ | 49.8↑ | 54.2 | 15.7↓ | 14.9↓ | 15.9↑ | | | | |
| | Medium-R | 38.4 | 36.5↓ | 24.6↓ | 15.7↓ | 55.1 | -2.6↓ | -2.4↑ | 4.8↑ | | | | |
| | Medium-E | 35.8 | 36.3↑ | 49.0↑ | 49.3↑ | 90.0 | 4.4↓ | 6.5↑ | 11.1↑ | | | | |

### A.3.5 Comparison with the cross-domain based baselines

In Tables 11 and 12, we provide the comparison between our DARA-based methods, fine-tune based methods, and MABE-based methods in Hopper and Walker2d tasks, over the dynamics shift concerning the joint noise of motion. We can observe that in a majority of tasks, our DARA-based methods outperforms the fine-tune-based method (67 ”↑” vs. 13 ”↓”, including the results in the main text). Moreover, our DARA can achieve comparable or better performance compared to MABE-based baselines on eleven out of sixteen tasks (including the results in the main text).

Table 11: Normalized scores in the (target) D4RL Hopper tasks with the joint noise shift., where ”Tune” denotes baseline ”fine-tune”.

| | Joint Noise Shift | Tune | DARA | Tune | DARA | Tune | DARA | Tune | DARA | Tune | DARA | $\pi_p \hat{T}$ | $\hat{T} \pi_p$ |
|---|---|---|---|---|---|---|---|---|---|---|---|---|---|
| | | BEAR | | BRAC-p | | BCQ | | CQL | | MOPO | | MABE | |
| Hopper | Random | 0.8 | 4.2↑ | 6.4 | 10.8↑ | 8.1 | 9.6↑ | **32.2** | 10.8↓ | 0.6 | 2.9↑ | 10.8 | 8.1 |
| | Medium | 1.9 | 2.0↑ | 44.9 | 37.6↓ | 47.7 | 54.4↑ | 52.5 | 58.0↑ | 0.8 | 17.3↑ | **63.5** | 57.7 |
| | Medium-R | 0.7 | 9.9↑ | 32.4 | **101.4**↑ | 29.6 | 32.0↑ | 1.3 | 3.6↑ | 1.8 | 6.4↑ | 21.5 | 35.4 |
| | Medium-E | 0.8 | 1.4↑ | 98.2 | 87.8↓ | 90.5 | **109.0**↑ | 107.3 | 108.9↑ | 4.9 | 7.5↑ | 15.5 | 104.8 |

Table 12: Normalized scores in the (target) D4RL Walker2d tasks with the joint noise shift., where ”Tune” denotes baseline ”fine-tune”.

| | Joint Noise Shift | Tune | DARA | Tune | DARA | Tune | DARA | Tune | DARA | Tune | DARA | $\pi_p \hat{T}$ | $\hat{T} \pi_p$ |
|---|---|---|---|---|---|---|---|---|---|---|---|---|---|
| | | BEAR | | BRAC-p | | BCQ | | CQL | | MOPO | | MABE | |
| Walker2d | Random | 2.7 | 2.6↓ | 1.4 | **8.8**↑ | 3.0 | 5.2↑ | 6.7 | 6.4↓ | -0.4 | -0.2↑ | 5.0 | -0.2 |
| | Medium | 0.5 | 0.1↓ | 55.8 | 72.9↑ | 45.1 | 52.7↑ | 76.6 | **81.2**↑ | 7.0 | 12.2↑ | 49.4 | 48.7 |
| | Medium-R | 3.2 | 10.4↑ | 12.2 | **34.8**↑ | 13.5 | 14.6↑ | -0.4 | 1.8↑ | 1.9 | 16.4↑ | 4.5 | 1.6 |
| | Medium-E | -0.4 | 0.6↑ | 71.7 | 74.3↑ | 44.8 | 57.2↑ | 104 | **116.5**↑ | 11.3 | 26.3↑ | 84.7 | 82.6 |

### A.3.6 Additional results on the quadruped robot

In this offline sim2real setting, we collect the source offline data in the simulator ($10^6$ or $2 * 10^6$ steps) and target offline data in the real world ($3 * 10^4$ steps). See Appendix A.4 for details. For testing, we directly deploy the learned policy in the real (flat or obstructive) environment and adopt the average distance covered in an episode (300 steps) as our evaluation metrics.

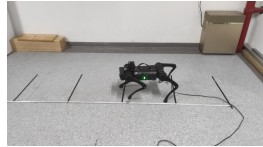 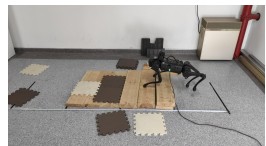

Figure 6: Illustration of the real environment (for testing): *(left)* the flat and static environment, *(right)* the obstructive and dynamic environment.

Table 13: Average distance (m) covered in an episode (300 steps) in flat and static (real) environment.

| Sim2real (Flat and Static) | | w/o Aug. | DARA | w/o Aug. | DARA | w/o Aug. | DARA |
|---|---|---|---|---|---|---|---|
| | | BCQ | | CQL | | MOPO | |
| Quadruped Robot | Medium | 1.56 | 1.64 ↑ | 1.80 | **1.82** ↑ | 0.00 | 0.00 |
| | Medium-R | 0.00 | 0.00 | — | — | — | — |
| | Medium-E | 2.16 | **2.47** ↑ | 2.03 | 2.02 ↓ | 0.00 | 0.00 |
| | Medium-R-E | 1.69 | **2.28** ↑ | 0.00 | 0.00 | — | — |
| Average performance improvement | | | 13.6% | | 0.2% | | 0.0% |

**(Flat and static environment)** We first deploy our learned policy in the flat and static environment. The results (distance covered in an episode) are provided in Table 13.

1) BCQ (Figure 7): We find that with Medium-R offline data, *w/o Aug. BCQ* and *DARA BCQ* both could not acquire the locomotion skills, which we think is caused by the lack of high-quality offline data. With more "expert" data (Medium-R → Medium → Medium-E, or Medium-R → Medium-R-E), *w/o-Aug. BCQ* allows for progressive performance (0.00 → 1.56 → 2.16, or 0.00 → 1.69 in BCQ), but with our reward augmentation, such performance can be further improved (with average improvement 13.6%).

2) CQL (Figure 8): We find that with Medium-R or Medium-R-E offline data, *w/o Aug. CQL* and *DARA CQL* both could not learn the locomotion skills, which we think is caused by the low-quality "Replay" offline data. With Medium or Medium-E offline data, *w/o Aug. CQL* and *DARA CQL* acquire similar performance on this flat and static environment.

3) MOPO: We find that the model-based MOPO (both *w/o Aug.* and *DARA*) could hardly learn the locomotion skill under the provided offline data.

Table 14: Average distance (m) covered in an episode (300 steps) in the obstructive and dynamic (real) environment.

| Sim2real (Obstructive and Dynamic) | | w/o Aug. | DARA | w/o Aug. | DARA | w/o Aug. | DARA |
|---|---|---|---|---|---|---|---|
| | | BCQ | | CQL | | MOPO | |
| Quadruped Robot | Medium | 0.85 | 1.35 ↑ | 0.92 | **1.40** ↑ | — | — |
| | Medium-R | — | — | — | — | — | — |
| | Medium-E | 1.15 | **1.41** ↑ | 0.77 | 1.32 ↑ | — | — |
| | Medium-R-E | 1.27 | **1.55** ↑ | — | — | — | — |
| Average performance improvement | | | 25.9% | | 30.9% | | — |

**(Obstructive and dynamic environment)** We then deploy our learned policy in the obstructive and dynamic environment. The results (distance covered in an episode) are provided in Table 14.

1) BCQ (Figure 9): In this obstructive environment, we can obtain similar results as in the flat environment. With more "expert" data (Medium → Medium-E → Medium-R-E), *w/o Aug. BCQ* allows for progressive performance (0.85 → 1.15 → 1.27), and with our reward augmentation, such performance can be further improved (with average improvement 25.9%). At the same time, we can also find that due to the presence of environmental obstacles, the performance of both *w/o*

*Aug. BCQ* and *w/o Aug. DARA* is decreased compared to the deployment on the flat environment. However, we find that our DARA exhibits greater average performance improvement under this obstructive environment ($13.6\% \rightarrow \textbf{25.9\%}$) compared to that in the flat environment. *These results demonstrate that our DARA can learn an adaptive policy for the target environment and thus show a greater advantage over* w/o-Aug. *in more complex environments.*

2) CQL (Figure 10): Similar to BCQ, our *DARA CQL* exhibits a greater performance improvement over baseline in the obstructive and dynamic environment ($0.2\% \rightarrow \textbf{30.9\%}$) compared to that in the flat and static environment.

**In summary**, The results in the quadruped robot tasks support our conclusion in the main text regarding the dynamics shift problem in offline RL — with only modest amounts of target offline data ($3 * 10^4$ steps), DARA-based methods can acquire an adaptive policy for the (both flat and obstructive) target environment and exhibit better performance compared to baselines under the dynamics adaptation setting.

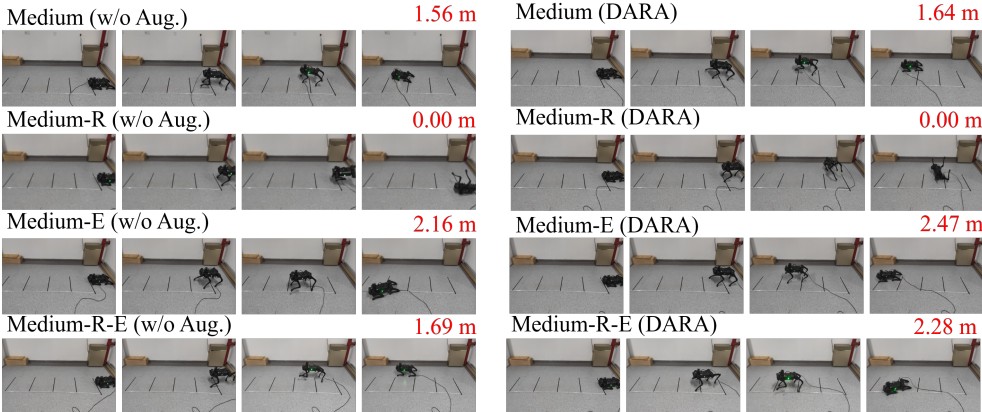

Figure 7: Deployment on the flat and static environment of BCQ.

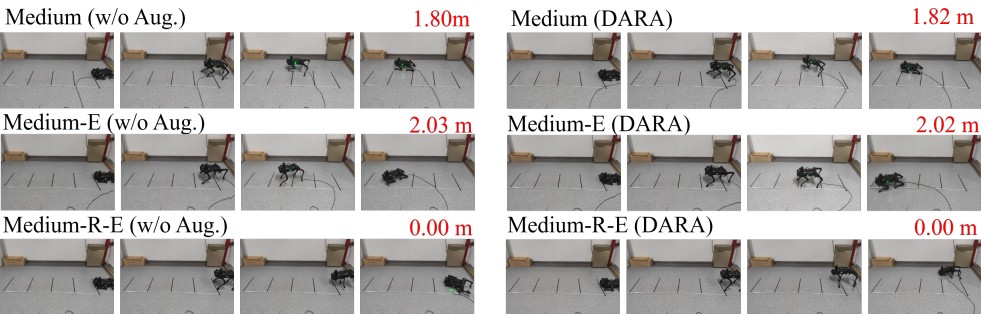

Figure 8: Deployment on the flat and static environment of CQL.

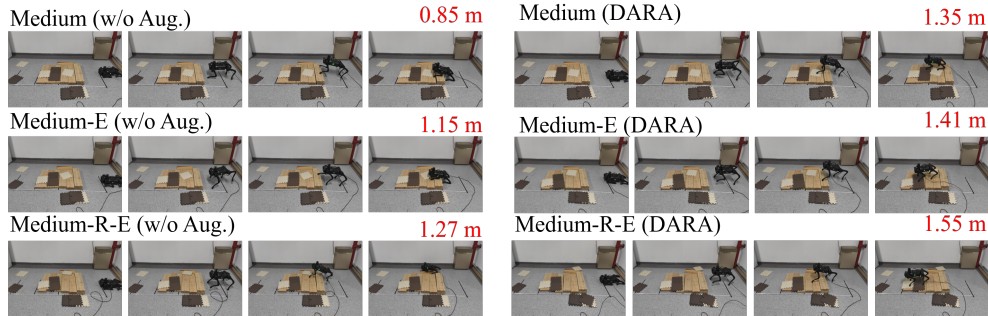

Figure 9: Deployment on the obstructive and dynamic environment of BCQ.

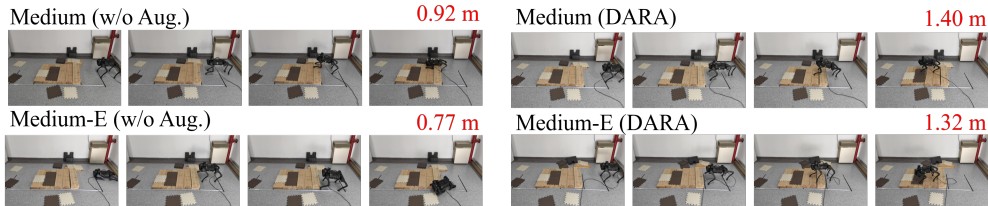

Figure 10: Deployment on the obstructive and dynamic environment of CQL.

### A.3.7  ABLATION STUDY WITH RESPECT TO THE AMOUNT OF TARGET OFFLINE DATA

To see whether the amount of target offline data can be further reduced, we show the results of the ablation study with respect to the amount of target offline data in Tables 15 and 16.

Table 15: Ablation study with respect to the amount of target Hopper data (body mass shift tasks). 10%, 5% and 1% denote training with 10%, 5% and 1% of target offline data, respectively, and additional 100% source offline data.

| Body Mass Shift | | 10% | 5% | 1% | 10% | 5% | 1% | 10% | 5% | 1% | 10% | 5% | 1% |
|---|---|---|---|---|---|---|---|---|---|---|---|---|---|
| | | | BEAR | | | BRAC-p | | | BCQ | | | CQL | |
| Hopper | Medium-R | 34.1 | 10.7 | 6.4 | 30.8 | 27.7 | 20.0 | 32.8 | 20.5 | 16.3 | 3.7 | 3.2 | 2.3 |
| | Medium-E | 1.2 | 0.6 | 0.6 | 34.7 | 25.1 | 20.6 | 84.2 | 65.1 | 55.6 | 99.7 | 52.3 | 38.5 |

Table 16: Ablation study with respect to the amount of target Walker2d data (body mass shift tasks). 10%, 5% and 1% denote training with 10%, 5% and 1% of target offline data, respectively, and additional 100% source offline data.

| Body Mass Shift | | 10% | 5% | 1% | 10% | 5% | 1% | 10% | 5% | 1% | 10% | 5% | 1% |
|---|---|---|---|---|---|---|---|---|---|---|---|---|---|
| | | | BEAR | | | BRAC-p | | | BCQ | | | CQL | |
| Walker2d | Medium-R | 7.3 | 5.9 | 1.3 | 18.6 | 21.6 | 15.8 | 15.1 | 12.7 | 9.7 | 2.0 | 1.3 | 0.5 |
| | Medium-E | 2.3 | -0.2 | -0.3 | 77.5 | 2.0 | -0.2 | 57.2 | 29.7 | 20.6 | 93.3 | 0.1 | -0.3 |

### A.3.8  ILLUSTRATION OF WHETHER THE LEARNED POLICY IS LIMITED TO THE SOURCE OFFLINE DATA

If we directly perform DARA with only the source offline data $\mathcal{D}'$, the learned behaviors will be restricted to the source offline data. For example, in the Map task, collecting source dataset with the obstacle and collecting target dataset without the obstacle. In this case, it can be harder for DARA (with only the source $\mathcal{D}'$) to capture the change in the transition dynamics, thus harder for the agent

to figure out the new optimal policy (the shorter path without the obstacle). However, as stated in Algorithm 1, we perform offline RL algorithms with both target offline data and source offline data $\{\mathcal{D}' \cup \mathcal{D}\}$. Thus, to some extent, such limitation can be overcome as long as offline RL algorithm captures the information (eg. the short path without the obstacle) contained in the (limited) target $\mathcal{D}$, see Figure 11 for the illustration.

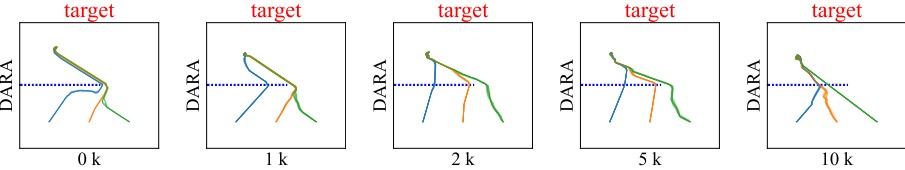

Figure 11: We exchange the source environment and the target environment in Figure 2 (in the main text) so that the source environment has an obstacle and the target environment has no obstacles. In the source domain, we collect 100k of random transitions. In the target domain, we collect 0k, 1k, 2k, 5k, and 10k random transitions respectively. We set $\eta = 0.1$. We can find that if we perform DARA with only source offline data $\mathcal{D}'$ (*i.e.*, 0k target data), we indeed can not acquire the optimal trajectory (eg. the short path without the obstacle). However, even there is no transition of passing through obstacles in the source data, performing DARA with $\{\mathcal{D}' \cup \mathcal{D}\}$ enables us to acquire the behavior of moving through obstacles. As we increase the number of target offline data $\mathcal{D}$, training with $\{\mathcal{D}' \cup \mathcal{D}\}$ can gradually acquire optimal trajectories.

### A.3.9 COMPARISON BETWEEN DARA AND IMPORTANCE SAMPLING (IS) BASED DYNAMICS CORRECTION

In Table 17, we report the experimental comparison between DARA and importance sampling based dynamics adaption. We can find that in most of the tasks, our DARA performs better than the IS-based approaches.

Table 17: Comparison between DARA and importance sampling (IS) based dynamics correction.

| Body Mass Shift | | IS | DARA | IS | DARA | IS | DARA |
|---|---|---|---|---|---|---|---|
| | | | BEAR | | BRAC-p | | AWR |
| Hopper | Random | $4.6 \pm 2.8$ | $\mathbf{8.4} \pm 1.2$ | $10.8 \pm 0.5$ | $\mathbf{11} \pm 0.6$ | $\mathbf{10.2} \pm 0.3$ | $4.5 \pm 0.9$ |
| | Medium | $1 \pm 0.4$ | $\mathbf{1.6} \pm 1$ | $17.4 \pm 10.6$ | $\mathbf{32.9} \pm 7.5$ | $24.8 \pm 7.7$ | $\mathbf{28.9} \pm 5.5$ |
| | Medium-R | $17.3 \pm 4.7$ | $\mathbf{34.1} \pm 5.8$ | $21.6 \pm 8.3$ | $\mathbf{30.8} \pm 4.9$ | $\mathbf{14} \pm 2.2$ | $4.2 \pm 3.5$ |
| | Medium-E | $0.8 \pm 0.2$ | $\mathbf{1.2} \pm 0.5$ | $\mathbf{36} \pm 13.5$ | $34.7 \pm 8.5$ | $\mathbf{29.3} \pm 2.6$ | $26.6 \pm 2$ |
| | | | BCQ | | CQL | | MOPO |
| Hopper | Random | $9.2 \pm 1.1$ | $\mathbf{9.7} \pm 0.2$ | $10.3 \pm 0.4$ | $\mathbf{10.4} \pm 0.4$ | $\mathbf{2.8} \pm 3$ | $2.1 \pm 1.7$ |
| | Medium | $28.2 \pm 8.8$ | $\mathbf{38.4} \pm 1.8$ | $43.3 \pm 10$ | $\mathbf{59.3} \pm 12.2$ | $7.6 \pm 7.2$ | $\mathbf{10.7} \pm 5.1$ |
| | Medium-R | $14.2 \pm 1.3$ | $\mathbf{32.8} \pm 0.9$ | $2.2 \pm 0.3$ | $\mathbf{3.7} \pm 1.4$ | $4.9 \pm 3.8$ | $\mathbf{8.4} \pm 3.5$ |
| | Medium-E | $83.4 \pm 23.7$ | $\mathbf{84.2} \pm 9.8$ | $87.8 \pm 16.9$ | $\mathbf{99.7} \pm 16.4$ | $4.6 \pm 2.9$ | $\mathbf{5.8} \pm 2.3$ |

### A.3.10 THE SENSITIVITY OF THE COEFFICIENT OF THE REWARD MODIFICATION

In Table 18, We check the sensitivity of hyper-parameter $\eta$, *i.e.*, the coefficient of the reward modification in $r(\mathbf{s}, \mathbf{a}) - \eta \Delta r(\mathbf{s}, \mathbf{a}, \mathbf{s}')$.

Table 18: We show the normalized scores for the Hopper tasks with body mass shift, by varying $\eta \in \{0, 0.05, 0.1, 0.2, 0.5\}$ over BEAR, BRAC-p, AWR, BCQ, CQL, and MOPO.

| Body Mass Shift | | Hyper-parameter $\eta$ | | | | |
|---|---|---|---|---|---|---|
| | | 0 | 0.05 | 0.1 | 0.2 | 0.5 |
| | | BEAR | | | | |
| Hopper | Random | $4.6 \pm 3.4$ | $7.7 \pm 0.9$ | $\mathbf{8.4} \pm 1.2$ | $7 \pm 1.2$ | $4.2 \pm 1.1$ |
| | Medium | $0.9 \pm 0.3$ | $1.1 \pm 0.6$ | $\mathbf{1.6} \pm 1$ | $0.9 \pm 0.2$ | $0.7 \pm 0.1$ |
| | Medium-R | $18.2 \pm 5$ | $28.5 \pm 5.9$ | $\mathbf{34.1} \pm 5.8$ | $29.1 \pm 4.4$ | $18.1 \pm 4.3$ |
| | Medium-E | $0.6 \pm 0$ | $0.8 \pm 0.1$ | $\mathbf{1.2} \pm 0.5$ | $\mathbf{1.2} \pm 0.6$ | $0.7 \pm 0.1$ |
| | | BRAC-p | | | | |
| Hopper | Random | $9.6 \pm 3.3$ | $\mathbf{11.2} \pm 0.8$ | $11 \pm 0.6$ | $10.6 \pm 2.4$ | $5.3 \pm 1.2$ |
| | Medium | $29.2 \pm 2.1$ | $26.5 \pm 1.8$ | $\mathbf{32.9} \pm 7.5$ | $16.1 \pm 0.9$ | $16.7 \pm 1.7$ |
| | Medium-R | $20.1 \pm 4.8$ | $17.8 \pm 3.2$ | $\mathbf{30.8} \pm 4.9$ | $13.9 \pm 1.7$ | $10.4 \pm 2.4$ |
| | Medium-E | $32.3 \pm 7.8$ | $\mathbf{40.4} \pm 4.4$ | $34.7 \pm 8.5$ | $29.4 \pm 6.5$ | $25.2 \pm 4.1$ |
| | | AWR | | | | |
| Hopper | Random | $3.4 \pm 0.7$ | $4.1 \pm 1$ | $\mathbf{4.5} \pm 0.9$ | $3.4 \pm 0.7$ | $2.5 \pm 0.1$ |
| | Medium | $20.8 \pm 6.3$ | $31.8 \pm 2.9$ | $\mathbf{28.9} \pm 5.5$ | $26.6 \pm 3.2$ | $17.4 \pm 1.5$ |
| | Medium-R | $4.1 \pm 1.7$ | $3 \pm 0.5$ | $4.2 \pm 3.5$ | $2.6 \pm 0.6$ | $\mathbf{4.3} \pm 1.3$ |
| | Medium-E | $26.8 \pm 0.4$ | $\mathbf{27} \pm 0$ | $26.6 \pm 2$ | $17.8 \pm 5.6$ | $24.2 \pm 3.9$ |
| | | BCQ | | | | |
| Hopper | Random | $8.3 \pm 0.3$ | $9.6 \pm 0.3$ | $\mathbf{9.7} \pm 0.2$ | $7.4 \pm 0.1$ | $7.6 \pm 0.3$ |
| | Medium | $25.7 \pm 5.5$ | $24.1 \pm 0.8$ | $\mathbf{38.4} \pm 1.8$ | $27.1 \pm 1.7$ | $26.7 \pm 0.8$ |
| | Medium-R | $28.7 \pm 1.9$ | $29.5 \pm 3$ | $\mathbf{32.8} \pm 0.9$ | $25.9 \pm 6$ | $21 \pm 2.2$ |
| | Medium-E | $75.4 \pm 7.8$ | $70.4 \pm 5.4$ | $\mathbf{84.2} \pm 9.8$ | $67.9 \pm 8$ | $61.9 \pm 4.3$ |
| | | CQL | | | | |
| Hopper | Random | $10.2 \pm 0.3$ | $10 \pm 0$ | $\mathbf{10.4} \pm 0.4$ | $10 \pm 0$ | $10 \pm 0$ |
| | Medium | $44.9 \pm 2.7$ | $\mathbf{59.8} \pm 6$ | $59.3 \pm 12.2$ | $44.2 \pm 1$ | $37.1 \pm 2.6$ |
| | Medium-R | $1.4 \pm 0.3$ | $2.1 \pm 0.2$ | $3.7 \pm 1.4$ | $\mathbf{3.9} \pm 1.7$ | $3.4 \pm 1$ |
| | Medium-E | $53.6 \pm 21.2$ | $65.3 \pm 15.4$ | $\mathbf{99.7} \pm 16.4$ | $60.5 \pm 16$ | $75.9 \pm 30$ |
| | | MOPO | | | | |
| Hopper | Random | $2 \pm 2.1$ | $1.8 \pm 0$ | $\mathbf{2.1} \pm 1.7$ | $1.2 \pm 0.4$ | $0.8 \pm 0$ |
| | Medium | $5 \pm 5.3$ | $6.5 \pm 1$ | $\mathbf{10.7} \pm 5.1$ | $5.3 \pm 1.6$ | $2.8 \pm 0.7$ |
| | Medium-R | $5.5 \pm 4.6$ | $7.5 \pm 0.8$ | $\mathbf{8.4} \pm 3.5$ | $5.7 \pm 3.5$ | $1.9 \pm 0.6$ |
| | Medium-E | $4.8 \pm 2.9$ | $\mathbf{8.1} \pm 1$ | $5.8 \pm 2.3$ | $4.7 \pm 0.7$ | $2.1 \pm 0.2$ |

## A.4 ENVIRONMENTS AND DATASET

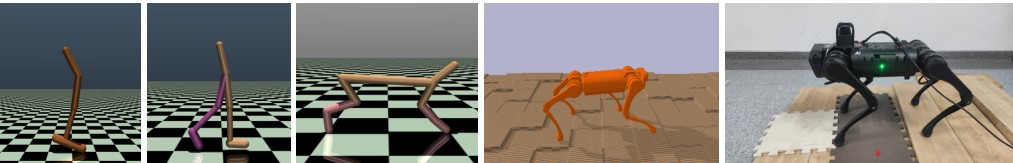

Figure 12: Illustration of the suite of tasks considered in this work: (from left to right) Hopper, Walker2d, Halfcheetah, simulated and real-world quadruped robots. These tasks require the RL agent to learn locomotion gaits for the illustrated characters.

In this work, the tasks include Hopper, Walker2d, HalfCheetah, simulated (see the dynamics parameters in Zhang et al.) and real-world quadruped robot, which are illustrated in Figure 12.

Table 19: Dynamics shift for Hopper, Walker2d, and Halfcheetah tasks. For the body mass shift, we change the mass of the body in the source MDP $M'$. For the joint noise shift, we add a noise (randomly sampling in $[-0.05, +0.05]$) to the actions when we collect the source offline data, *i.e.*, $\mathcal{D}' := \{(\mathbf{s}, \mathbf{a}, r, \mathbf{s}')\} \sim d_{\mathcal{D}'}(\mathbf{s})\pi_{b'}(\mathbf{a}|\mathbf{s})r(\mathbf{s}, \mathbf{a})T'(\mathbf{s}'|\mathbf{s}, \mathbf{a} + \text{noise})$.

|  | Hopper | | Walker2d | | HalfCheetah | |
|---|---|---|---|---|---|---|
|  | Body Mass Shfit | Joint Noise Shift | Body Mass Shfit | Joint Noise Shift | Body Mass Shfit | Joint Noise Shift |
| Source | mass[-1]=2.5 | action[-1]+noise | mass[-1]=1.47 | action[-1]+noise | mass[4]=0.5 | action[-1]+noise |
| Target | mass[-1]=5.0 | action[-1]+0 | mass[-1]=2.94 | action[-1]+0 | mass[4]=1.0 | action[-1]+0 |

In the Hopper, Walker2d and HalfCheetah dynamics adaptation setting, we set the D4RL (Fu et al., 2020) dataset as our target domain. For the source dynamics, we change the body mass (body mass shift) or add noises to joints (joint noise shift) of the agents (see Table 19 for the details) and then collect the source offline dataset in the changed environment. Following Fu et al. (2020), on the changed source environment, we collect the 1) "Random" offline data, generated by unrolling a randomly initialized policy, 2) "Medium" offline data, generated by a trained policy with the "medium" level of performance in the source environment, 3) "Medium-Replay" (Medium-R) offline data, consisting of recording all samples in the replay buffer observed during training until the policy reaches the "medium" level of performance, 4) "Medium-Expert" (Medium-E) offline data, mixing equal amounts of expert demonstrations and "medium" data in the source environment.

In the sim2real setting (for the quadruped robot), we use the A1 dog from Unitree (Wang, 2020). We collect the target offline data using five target behavior policies in the real-world with changing terrains, as shown in Figure 13, and collect the "Medium", "Medium-Replay" (Medium-R), "Medium-Expert" (Medium-E), "Medium-Replay-Expert" (Medium-R-E) source offline data in the simulator, where "Medium-Replay-Expert" denotes mixing equal amounts of "Medium-Replay" data and expert demonstrations in the simulator. In Section A.5, we provide the details of how to obtain the target and source behavior policy, so as to collect our target and source offline data.

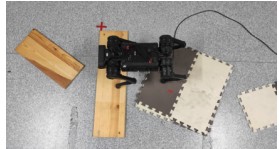

Figure 13: Real-world terrains (for collecting the target offline data).

We list our tasks properties in Table 20 and provide our collected dataset in supplementary material. In implementation, we set $\eta = 0.1$ for all simulated tasks and set $\eta = 0.01$ for the sim2real task. In Table 18, we also report the sensitivity of DARA on the hyper-parameters $\eta$.

## A.5 TRAINING THE (TARGET AND SOURCE) BEHAVIOR POLICY FOR THE QUADRUPED ROBOT

To obtain a behavior policy that can be deployed in simulator (for collecting the source offline data) or real-world (for collecting the target offline data), we introduce the prior knowledge (Iscen et al., 2018) and domain randomization (Tobin et al., 2017; Peng et al., 2018).

Table 20: Statistics for each task in our adaptation setting.

| Environment | Dynamics Shift | Task Name | Target (1T) | Source (10S) |
|---|---|---|---|---|
| Hopper | Body Mass Shfit | Random | $10^5$ (D4RL) | $10^6$ |
| | | Medium | $10^5$ (D4RL) | $10^6$ |
| | | Medium-Replay | 20092 (D4RL) | $10^6$ |
| | | Medium-Expert | $2 * 10^5$ (D4RL) | $2 * 10^6$ |
| | Joint Noise Shift | Random | $10^5$ (D4RL) | $10^6$ |
| | | Medium | $10^5$ (D4RL) | $10^6$ |
| | | Medium-Replay | 20092 (D4RL) | $10^6$ |
| | | Medium-Expert | $2 * 10^5$ (D4RL) | $2 * 10^6$ |
| Walker2d | Body Mass Shfit | Random | $10^5$ (D4RL) | $10^6$ |
| | | Medium | $10^5$ (D4RL) | $10^6$ |
| | | Medium-Replay | 10093 (D4RL) | $10^6$ |
| | | Medium-Expert | $2 * 10^5$ (D4RL) | $2 * 10^6$ |
| | Joint Noise Shift | Random | $10^5$ (D4RL) | $10^6$ |
| | | Medium | $10^5$ (D4RL) | $10^6$ |
| | | Medium-Replay | 10093 (D4RL) | $10^6$ |
| | | Medium-Expert | $2 * 10^5$ (D4RL) | $2 * 10^6$ |
| HalfCheetah | Body Mass Shfit | Random | $10^5$ (D4RL) | $10^6$ |
| | | Medium | $10^5$ (D4RL) | $10^6$ |
| | | Medium-Replay | 10100 (D4RL) | $10^6$ |
| | | Medium-Expert | $2 * 10^5$ (D4RL) | $2 * 10^6$ |
| | Joint Noise Shift | Random | $10^5$ (D4RL) | $10^6$ |
| | | Medium | $10^5$ (D4RL) | $10^6$ |
| | | Medium-Replay | 10100 (D4RL) | $10^6$ |
| | | Medium-Expert | $2 * 10^5$ (D4RL) | $2 * 10^6$ |
| A1 robot (Unitree) | Sim2Real | Medium | $3 * 10^4$ (real-world) | $10^6$ (simulator) |
| | | Medium-Replay | $3 * 10^4$ (real-world) | $10^6$ (simulator) |
| | | Medium-Expert | $3 * 10^4$ (real-world) | $2 * 10^6$ (simulator) |
| | | Medium-Replay-Expert | $3 * 10^4$ (real-world) | $2 * 10^6$ (simulator) |

**Prior Knowledge:** To reduce the impact of the foot at the moment of touching the ground during the robot locomotion, we designed a compound cycloid trajectory (Sakakibara et al., 1990) as prior knowledge. In our implementation for the foot trajectory, four aspects are mainly considered: 1) The robot walks stably without obvious shaking; 2) The joint impact of the robot during the locomotion is small; 3) The joint speed and acceleration of the robot during the locomotion are continuous and smooth; 4) The feet of the robot will not slide when they are in contact with the ground. Similar to Lee et al. (2020), we define a periodic phase variable $\phi_i \in [0.0, 0.6], i = 1, 2, 3, 4$ for each leg, which represents swing phase if $\phi_i \in [0.0, 0.3]$ and contact phase if $\phi_i \in [0.3, 0.6]$. At every time step $t$, $\phi_i = (t * f_0 + \phi_0[i] + \phi_{\text{offset}}[i])(\text{mod } 2T_m)$ where $T_m = 0.3$, and $f_0 = 1.1$ is the base frequency, and $\phi_0 = [0, 0.3, 0.3, 0]$ is the initial phase. $\phi_{\text{offset}}$ is part of the output of the controller. The trajectory of the swing leg is:

$$\begin{cases} x_i = S \left[ \frac{t}{T_m} - \frac{1}{2\pi} \sin\left(\frac{2\pi t}{T_m}\right) \right] + S_0, i = 1, 2 \\ x_i = S \left[ \frac{t}{T_m} - \frac{1}{2\pi} \sin\left(\frac{2\pi t}{T_m}\right) \right] - S + S_0, i = 3, 4 \\ y = Y_0 \\ z = H \left[ \text{sgn}\left(\frac{T_m}{2} - t\right)(2f_E(t) - 1) + 1 \right] + Z_0 \end{cases},$$

where

$$f_E(t) = \frac{t}{T_m} - \frac{1}{4\pi} \sin\left(\frac{4\pi t}{T_m}\right),$$

and

$$\text{sgn}\left(\frac{T_m}{2} - t\right) = \begin{cases} 1 & 0 \leq t < \frac{T_m}{2} \\ -1 & \frac{T_m}{2} \leq t < T_m \end{cases}.$$

The trajectory of the standing leg is:

$$
\begin{cases}
x_i = S\left(\frac{2T_m - t}{T_m} + \frac{1}{2\pi}\sin\left(\frac{2\pi t}{T_m}\right)\right) + S_0, i = 1, 2 \\
x_i = S\left(\frac{2T_m - t}{T_m} + \frac{1}{2\pi}\sin\left(\frac{2\pi t}{T_m}\right)\right) - S + S_0, i = 3, 4 \\
y = Y_0 \\
z = Z_0
\end{cases}.
$$

where $S = 0.14m, H = 0.18m$ are the maximum foot length and height. $S_0 = [0.17, 0.17, -0.2, -0.2], Y_0 = [-0.13, 0.13, -0.13, 0.13], Z_0 = [-0.32, -0.32, -0.32, -0.32]$ are the default target foot position in body frame.

**Domain Randomization:** To encourage the policy to be robust to variations in the dynamics, we incorporate the domain randomization. In Table 21, we provide the dynamics parameters and their respective range of values.

Table 21: Dynamic parameters and their respective range of values utilized during training.

| Parameter | Range |
| --- | --- |
| Mass | $[0.95, 1.1]$ ×default value |
| Inertia | $[0.80, 1.2]$ ×default value |
| Motor Strength | $[0.80, 1.2]$ ×default value |
| Latency | $[0, 0.04]$ $s$ |
| Lateral Friction | $[0.5, 1.25]$ $Ns/m$ |
| Joint Friction | $[0, 0.05]$ $Nm$ |

**State Space, Action Space and Reward Function:** The action is a 16-dimensional vector consisting of leg phase and target foot position residuals in the body frame. The design of state space and reward function mainly follows the prior work Lee et al. (2020). In Table 22, we provide the state representation.

Table 22: State representation for the behavior policy.

| Data | Dimension |
| --- | --- |
| Desired direction$\left(\left(^B_{IB}\hat{v}_d\right)_{xy}\right)$ | 2 |
| Euler angle$(rpy)$ | 3 |
| Base angular velocity$\left(^B_{IB}\omega\right)$ | 3 |
| Base linear velocity$\left(^B_{IB}v\right)$ | 3 |
| Joint position/velocity$\left(\theta_i, \dot{\theta}_i\right)$ | 24 |
| FTG phases$(\sin(\phi_i), \cos(\phi_i))$ | 8 |
| FTG frequencies$(f_i)$ | 4 |
| Base frequency$(f_o)$ | 1 |
| Joint position error history | 24 |
| Joint velocity history | 24 |
| Foot target history$\left(\left(r_{f,d}\right)_{t-1,t-2}\right)$ | 24 |

The reward function is defined as

$$
0.1r_{lv} + 0.05r_y + 0.05r_{rp} + 0.005r_b + 0.02r_{bc} + 0.025r_s + 2\cdot 10^{-5}r_\tau.
$$

The individual terms are defined as follows.

1) Linear velocity reward $r_{lv}$ :

$$
r_{lv} := \begin{cases} exp(-30|v_{pr} - 0.2|) & v_{pr} < 0.2 \\ 1 & v_{pr} \geq 0.2 \end{cases},
$$

where $v_{pr} = v_{xy} \cdot \hat{v}_{xy}$ is the base linear velocity projected onto the command direction.

2) Yaw angle reward $r_y$ :

$$r_y := exp(-(y - \hat{y})^2), \tag{17}$$

where $y$ and $\hat{y}$ is the yaw and desired yaw angle.

3) Roll and pitch reward $r_{rp}$ :

$$r_{rp} := exp(-1.5 \sum (\phi - [0, \arccos(\frac{< P_{xz}, (0, 0, 1)^T >}{\| P_{xz} \|}) - \pi/2])^2), \tag{18}$$

where $\phi$ are the roll and pitch angle. $P_{xz} = P_1 - P_4$ or $P_{xz} = P_2 - P_3$, $P_i, i \in [1, 4]$ are the foot position in world frame. The advantage of designing the target pitch angle in this way is to ensure that the body of the robot is parallel to the supporting surface of the stand legs, thereby ensuring that the robot can smoothly over challenge terrain, such as upward stairs.

4) Base motion reward $r_b$ :

$$r_b := exp(-1.5 \sum (v_{xy} - v_{pr} * \hat{v}_{xy})^2) + exp(-1.5 \sum (\omega_{xy})^2), \tag{19}$$

where $\omega_{xy}$ are the roll and pitch rates.

5) Body collision reward $r_{bc}$ :

$$r_{bc} := -|I_{body}/I_{foot}|, \tag{20}$$

where $I_{body}$ and $I_{foot}$ are the contact numbers of robot's body parts and foot with the terrain, respectively.

6) Target smooth reward $r_s$ :

$$r_s := -||f_{d,t} - 2f_{d,t-1} + f_{d,t-2}||, \tag{21}$$

where $f_{d,i}(i = t, t - 1, t - 2)$ are the target foot positions in the time-step $t$, $t - 1$ and $t - 2$.

7) Torqure reward $r_\tau$:

$$r_\tau := - \sum_i |\tau_i|, \tag{22}$$

where $\tau_i$ is the joint torques.

**Training Details:** Both the behavior policy and value networks are Multilayer Perceptron (MLP) with 3 hidden layers, which have 256, 128 and 64 nodes. The activation function is the *Tanh* function, and the optimizer is *Adam*. With the above prior knowledge, domain randomization and reward function, we train our behavior policy with SAC (Haarnoja et al., 2018) in PyBullet (Coumans & Bai, 2016–2021).

## A.6 ADDITIONAL RESULTS

Here we provide additional results regarding the error bars (Tables 23 and 24).

Table 23: Normalized scores for the D4RL tasks (with body mass shift). We take the baseline results (for 10T) of MOPO from their original papers and that of the other model-free methods (BEAR, BRAC-p, AWR, BCQ and CQL) from the D4RL paper (Fu et al., 2020).

| | | BEAR | | | | BRAC-p | | | | AWR | | | |
|---|---|---|---|---|---|---|---|---|---|---|---|---|---|
| | Body Mass Shift | 10T | 1T | 1T+10S w/o Aug. | 1T+10S (DARA) | 10T | 1T | 1T+10S w/o Aug. | 1T+10S (DARA) | 10T | 1T | 1T+10S w/o Aug. | 1T+10S (DARA) |
| Hopper | Random | 11.4 | 1 ± 0.5 | 4.6 ± 3.4 | 8.4 ± 1.2 | 11 | 10.9 ± 0.1 | 9.6 ± 3.3 | 11 ± 0.6 | 10.2 | 10.3 ± 0.3 | 3.4 ± 0.7 | 4.5 ± 0.9 |
| Hopper | Medium | 52.1 | 0.8 ± 0 | 0.9 ± 0.3 | 1.6 ± 1 | 32.7 | 29 ± 6.2 | 29.2 ± 2.1 | 32.9 ± 7.5 | 35.9 | 30.9 ± 0.4 | 20.8 ± 6.3 | 28.9 ± 5.5 |
| Hopper | Medium-R | 33.7 | 1.3 ± 1.5 | 18.2 ± 5 | 34.1 ± 5.8 | 0.6 | 5.4 ± 3.3 | 20.1 ± 4.8 | 30.8 ± 4.9 | 28.4 | 8.8 ± 4.9 | 4.1 ± 1.7 | 4.2 ± 3.5 |
| Hopper | Medium-E | 96.3 | 0.8 ± 0.1 | 0.6 ± 0 | 1.2 ± 0.5 | 1.9 | 34.5 ± 14.7 | 32.3 ± 7.8 | 34.7 ± 8.5 | 27.1 | 27 ± 1.3 | 26.8 ± 0.4 | 26.6 ± 2 |
| Walker2d | Random | 7.3 | 1.5 ± 0.9 | 3.1 ± 0.9 | 3.2 ± 0.4 | -0.2 | 0 ± 0.2 | 1.3 ± 0.7 | 3.2 ± 2.5 | 1.5 | 1.3 ± 0.4 | 2 ± 1 | 2.4 ± 0.8 |
| Walker2d | Medium | 59.1 | -0.5 ± 0.3 | 0.6 ± 0.5 | 0.3 ± 0.7 | 77.5 | 6.4 ± 9.9 | 70 ± 10.1 | 78 ± 3.1 | 17.4 | 14.8 ± 2.8 | 17.1 ± 0.2 | 17.2 ± 0.1 |
| Walker2d | Medium-R | 19.2 | 0.7 ± 0.6 | 6.5 ± 5.1 | 7.3 ± 1.3 | -0.3 | 8.5 ± 2.2 | 9.9 ± 2 | 18.6 ± 6.5 | 15.5 | 7.4 ± 2.1 | 1.6 ± 0.4 | 1.5 ± 0.3 |
| Walker2d | Medium-E | 40.1 | -0.1 ± 0.1 | 1.5 ± 2.5 | 2.3 ± 2.2 | 76.9 | 20.6 ± 16.8 | 64.1 ± 10.8 | 77.5 ± 3.1 | 53.8 | 35.5 ± 10.4 | 52.5 ± 1.2 | 53.3 ± 0.3 |

| | | BCQ | | | | CQL | | | | MOPO | | | |
|---|---|---|---|---|---|---|---|---|---|---|---|---|---|
| | Body Mass Shift | 10T | 1T | 1T+10S w/o Aug. | 1T+10S (DARA) | 10T | 1T | 1T+10S w/o Aug. | 1T+10S (DARA) | 10T | 1T | 1T+10S w/o Aug. | 1T+10S (DARA) |
| Hopper | Random | 10.6 | 10.6 ± 0.1 | 8.3 ± 0.3 | 9.7 ± 0.2 | 10.8 | 10.6 ± 0.1 | 10.2 ± 0.3 | 10.4 ± 0.4 | 11.7 | 4.8 ± 2.4 | 2 ± 2.1 | 2.1 ± 1.7 |
| Hopper | Medium | 54.5 | 37.1 ± 6.3 | 25.7 ± 5.5 | 38.4 ± 1.8 | 58 | 43 ± 9.2 | 44.9 ± 2.7 | 59.3 ± 12.2 | 28 | 4.1 ± 2 | 5 ± 5.3 | 10.7 ± 5.1 |
| Hopper | Medium-R | 33.1 | 9.3 ± 4.4 | 28.7 ± 1.9 | 32.8 ± 0.9 | 48.6 | 9.6 ± 5.2 | 1.4 ± 0.3 | 3.7 ± 1.4 | 67.5 | 1 ± 0.6 | 5.5 ± 4.6 | 8.4 ± 3.5 |
| Hopper | Medium-E | 110.9 | 58 ± 16.2 | 75.4 ± 7.8 | 84.2 ± 9.8 | 98.7 | 59.7 ± 34.5 | 53.6 ± 21.2 | 99.7 ± 16.4 | 23.7 | 1.6 ± 0.6 | 4.8 ± 2.9 | 5.8 ± 2.3 |
| Walker2d | Random | 4.9 | 1.8 ± 0.9 | 4.5 ± 0.5 | 4.8 ± 0.3 | 7 | 1.7 ± 1.3 | 3.2 ± 1.4 | 3.4 ± 1.9 | 13.6 | -0.2 ± 0.2 | -0.1 ± 0.1 | -0.1 ± 0.2 |
| Walker2d | Medium | 53.1 | 32.8 ± 8.2 | 50.9 ± 4.3 | 52.3 ± 1.4 | 79.2 | 42.9 ± 24.2 | 80 ± 1.2 | 81.7 ± 3.1 | 17.8 | 7 ± 3.6 | 5.7 ± 4.7 | 11 ± 4.3 |
| Walker2d | Medium-R | 15 | 6.9 ± 0.6 | 14.9 ± 0.2 | 15.1 ± 0.2 | 26.7 | 4.6 ± 3.9 | 0.8 ± 0.5 | 2 ± 1.5 | 39 | 5.1 ± 5.7 | 3.1 ± 2.4 | 14.2 ± 4.5 |
| Walker2d | Medium-E | 57.5 | 32.5 ± 9.1 | 55.2 ± 3.8 | 57.2 ± 0.2 | 111 | 49.5 ± 26.7 | 63.5 ± 22.5 | 93.3 ± 8.8 | 44.6 | 5.3 ± 3.9 | 5.5 ± 3.5 | 17.2 ± 8.7 |

Table 24: Normalized scores for the D4RL tasks (with joint noise shift). We take the baseline results (for 10T) of MOPO from their original papers and that of the other model-free methods (BEAR, BRAC-p, AWR, BCQ and CQL) from the D4RL paper (Fu et al., 2020).

| Joint Noise Shift | 10T | 1T | 1T+10S w/o Aug. | 1T+10S (DARA) | 10T | 1T | 1T+10S w/o Aug. | 1T+10S (DARA) | 10T | 1T | 1T+10S w/o Aug. | 1T+10S (DARA) |
|---|---|---|---|---|---|---|---|---|---|---|---|---|
| **Hopper** — BEAR / BRAC-p / AWR | | | | | | | | | | | | |
| Random | 11.4 | 0.6 ± 0 | 7.4 ± 0.5 | 4.2 ± 3.6 | 11 | 10.8 ± 0.2 | 10 ± 0.8 | 10.8 ± 0 | 10.2 | 10.1 ± 0 | 3.6 ± 0 | 4 ± 0.4 |
| Medium | 52.1 | 0.8 ± 0 | 2 ± 1 | 2 ± 0.1 | 32.7 | 26.6 ± 4.8 | 27.6 ± 2.8 | 37.6 ± 7 | 35.9 | 30.3 ± 0.1 | 38.8 ± 3.9 | 41.3 ± 5.4 |
| Medium-R | 33.7 | 2.7 ± 1.6 | 3.6 ± 0.4 | 9.9 ± 6.3 | 0.6 | 13.4 ± 6.4 | 89.9 ± 7.8 | 101.4 ± 0.2 | 28.4 | 12.4 ± 6 | 6.7 ± 4.2 | 7.2 ± 0.2 |
| Medium-E | 96.3 | 0.8 ± 0.2 | 0.8 ± 0 | 1.4 ± 0.6 | 1.9 | 19.8 ± 14 | 57.6 ± 23.4 | 87.8 ± 13.3 | 27.1 | 25.5 ± 1.2 | 27 ± 0 | 27 ± 0.1 |
| **Hopper** — BCQ / CQL / MOPO | | | | | | | | | | | | |
| Random | 10.6 | 10.5 ± 0.1 | 7 ± 0 | 9.6 ± 0 | 10.8 | 10.4 ± 0.1 | 10.4 ± 0.4 | 10.8 ± 0 | 11.7 | 1.5 ± 0.8 | 1.3 ± 0.5 | 2.9 ± 1.5 |
| Medium | 54.5 | 45.8 ± 2.2 | 49 ± 1.7 | 54.4 ± 0.1 | 58 | 46.2 ± 11.9 | 58 ± 0 | 58 ± 0 | 28 | 2.7 ± 2.1 | 9.2 ± 5.4 | 17.3 ± 3.4 |
| Medium-R | 33.1 | 13 ± 5 | 23.8 ± 3.2 | 32 ± 0.9 | 48.6 | 13.6 ± 6.4 | 2.6 ± 0.3 | 3.6 ± 0.6 | 67.5 | 0.8 ± 0.1 | 2.3 ± 1.7 | 6.4 ± 0.8 |
| Medium-E | 110.9 | 44.6 ± 18.6 | 96 ± 0.5 | 109 ± 0.2 | 98.7 | 50.7 ± 26.9 | 73.4 ± 1.5 | 108.9 ± 0.7 | 23.7 | 1 ± 0.2 | 6.1 ± 1.4 | 7.5 ± 0.6 |
| **Walker2d** — BEAR / BRAC-p / AWR | | | | | | | | | | | | |
| Random | 7.3 | 2.2 ± 0.1 | 0.6 ± 0.1 | 2.6 ± 0.3 | -0.2 | 2.8 ± 2.8 | 3.3 ± 2.9 | 8.8 ± 8 | 1.5 | 0.9 ± 0.1 | 1.5 ± 0.6 | 1.5 ± 0.2 |
| Medium | 59.1 | -0.4 ± 0.1 | 0.6 ± 0 | 0.1 ± 0.3 | 77.5 | 28.8 ± 28.4 | 55.2 ± 15.8 | 72.9 ± 9.1 | 17.4 | 12.2 ± 0.3 | 17.2 ± 0.2 | 17.2 ± 0.2 |
| Medium-R | 19.2 | 0.4 ± 0.2 | 4 ± 0.2 | 10.4 ± 2.4 | -0.3 | 6.3 ± 1.2 | 32.1 ± 11.9 | 34.8 ± 10.5 | 15.5 | 6 ± 1 | 1.4 ± 0 | 2.1 ± 0.9 |
| Medium-E | 40.1 | -0.2 ± 0.2 | 0.8 ± 0.4 | 0.6 ± 0.6 | 76.9 | 21.8 ± 18.4 | 62.3 ± 13.1 | 74.3 ± 1.8 | 53.8 | 40.4 ± 12.6 | 53 ± 0.1 | 53.6 ± 0 |
| **Walker2d** — BCQ / CQL / MOPO | | | | | | | | | | | | |
| Random | 4.9 | 3.7 ± 1.8 | 3.4 ± 0.4 | 5.2 ± 0.3 | 7 | 0.5 ± 1 | 2.7 ± 0.2 | 6.4 ± 0.6 | 13.6 | -0.3 ± 0.1 | -0.2 ± 0 | -0.2 ± 0.2 |
| Medium | 53.1 | 43 ± 8.3 | 44.9 ± 3.3 | 52.7 ± 0.3 | 79.2 | 43.9 ± 21.7 | 73.2 ± 0.8 | 81.2 ± 1.1 | 17.8 | 5.8 ± 5.9 | 7.8 ± 6.2 | 12.2 ± 5 |
| Medium-R | 15 | 5.7 ± 0.5 | 9.8 ± 5.2 | 14.6 ± 0.4 | 26.7 | 1.8 ± 1.2 | 1.4 ± 0.4 | 1.8 ± 0.4 | 39 | 0.8 ± 0.7 | 9.3 ± 5.2 | 16.4 ± 4.9 |
| Medium-E | 57.5 | 44.5 ± 3.6 | 40.6 ± 16.4 | 57.2 ± 0.2 | 111 | 46.8 ± 40 | 109.9 ± 4.5 | 116.5 ± 9.1 | 44.6 | 2.9 ± 3.1 | 15.2 ± 12.8 | 26.3 ± 18.4 |

