# OpenReview forum: "DARA: Dynamics-Aware Reward Augmentation in Offline Reinforcement Learning"
_ICLR.cc/2022/Conference — ICLR 2022 Poster_

### Official Review · Reviewer_i5hn · 2021-10-24

**Correctness:** 3
**Technical Novelty And Significance:** 2
**Empirical Novelty And Significance:** 2
**Recommendation:** 6
**Confidence:** 3

**Main Review:**


Strength:

* Overall, the paper attempts to answer important question in offline reinforcement learning.
* The given empirical evaluation seems theoretically grounded, results seem reasonably significant, with improved performance over the baseline source dataset.

Weakness:
* The motivation of the proposed approach needs to be expanded further. There is a single line which is hard to fully understand. "Motivated by the (approximate) equivalence1 between the dynamics change and the rewards mod- ification (Kumar et al., 2020b; Eysenbach & Levine, 2019; Eysenbach et al., 2021), we propose to modify rewards in the source offline data to compensate for the dynamics shift. "


* Clarity of writing is a major weakness of the paper. It's not very well organized, logical flow is missing. As one example, the "Related Work"(Section 5) is towards the end of the paper, after the approach has been extensively described. A few more highlighted below to fix:

* "This modification aims to discourage the learning from these source offline transitions (next-state) that are hard to be achieved in the     target environment": Can the latter part of the statement be more precise? This could be misinterpreted as a goat state to be.

* Algorithm 1 Step 1: Perhaps clarify both the models are learnt independently for each of the datasets, ie 4 models.

* Section 4.1: What does it mean for a trajectory to be "optimal"? In reality, is this actually binary?

* Section 4.1 Figure 2 isn't clear to understand

* Section 6.1 There is no dashed green line in Figure 3, as mentioned in Section 6.1

* Section Was the source(noisy) dataset also collected from a D4RL policy?

* "Specifically, we out- line state-action-next-state: besides characterizing the state-action distribution shift, we additionally penalize the agent with a dynamics-aware reward modification.": This line isn't very clear to me.

* Section 4.1: Typo: "the limit"


**Summary Of The Paper:**

The given work proposes the addition of a dynamics aware framework for Offline Reinforcement Learning. It motivates that  offline reinforcement learning suffer from a dynamics shift in SARSA transitions, which need to be effectively handled. They propose modifying reward in source offline dataset, penalizing it with the dynamics aware reward transition, setting the lower bound. The reward augmentation is done with explicit policy constraints as well as implicit policy constraints.They evaluate this approach on various robotic manipulation task from the Gym-MujoCo suite, evaluating and showcasing perforomance on four offline RL algorithms.



**Summary Of The Review:**

Overall, the given work is theoretically grounded, providing incremental improvement over the past work in this area of tackling distribution shifts for offline RL learning. The results seem promising. However, the paper is not the easiest to read and interpret.  I'm willing to reconsider my decision, based on the feedback addressed by the authors.

Edit: Updated score based on reviewer feedback

---

> ### Author Response · Authors · 2021-11-15
> **Author Response**
>
> Thank you very much for your feedback. We really appreciate the concerns and questions brought up. Please see our response below.
>
> (1) **Motivation** We have changed the exposition of our motivation in "Introduction" section.
>
> (2) **Related Work** We have moved the "Related Work" section to the front of the "Preliminaries" section.
>
> (3) **Statement** We have changed the exposition (in "Introduction" section) mentioned by the reviewer to "that are likely in the source environment but are unlikely in the target environment".
>
> (4) **Algorithm 1 Step 1** We have changed the exposition of Step 1 in Algorithm 1 to "Learn classifiers ($q_{\text{sas}}$ and $q_{\text{sa}}$) that distinguish source-domain transition from target-domain transition".
>
> (5) **Optimal** An optimal trajectory means that the actions in each state of the trajectory are optimal. In reality, a trajectory is not necessarily optimal. However, in RL, we usually pursue the optimal policy or the corresponding (optimal) trajectory. Actually, the "optimal" term comes from the control-as-inference framework [1, 2], where the control (RL) problem is simply an inference problem in a particular type of probabilistic graphical model. In the probabilistic graphical model, we thus maximize the likelihood of the "optimal" trajectory, $p(\mathcal{O}=1|\tau)=\exp ( \sum_t r_t/\eta)$, which is consistent with seeking an "optimal" policy in the control (RL) problem.
>
> [1] Abdolmaleki, Abbas, et al. "Maximum a posteriori policy optimisation." (2018). \\\
> [2] Levine, Sergey. "Reinforcement learning and control as probabilistic inference: Tutorial and review." (2018).
>
> (6) **Figure 2** We have added more description w.r.t. Figure 2.
>
> (7) **Green** In Section 6.1, "dashed green line" should be "dashed blue line".
>
> (8) **D4RL policy** Sorry that we are not quite sure what a D4RL policy (mentioned by the reviewer) is. It seems that D4RL does not provide any (behavior) policy. Our source offline data is generated by first training a (behavior) policy from scratch on the source environment, and collecting samples from this trained (behavior) policy on the same source environment. We have further elaborated the details in the appendix.
>
> (9) **Outline** We have changed the exposition mentioned by the reviewer. We refer the reviewer to our revised paper for the details.
>
> (10) **Typo** We have changed the typo "the limit".
>
> We hope the above response addresses the reviewer’s concerns and welcome any further feedback.

---

> ### Author Response · Authors · 2021-11-18
> **Response to "Edit: Updated score based on reviewer feedback"**
>
> Thank you for the response! We really appreciate your comments!

---

### Official Review · Reviewer_QyWv · 2021-11-02

**Correctness:** 3
**Technical Novelty And Significance:** 4
**Empirical Novelty And Significance:** 3
**Recommendation:** 6
**Confidence:** 3

**Main Review:**

The paper shows several strengths:

(+) The method proposed in the paper contributes to improving the data efficiency in the offline reinforcement learning setting. It both reduces the amount of data needed from the target environment and relaxes the requirement on the reward information contained in the target environment’s dataset.

(+) The theory behind the idea is well explained. The blue and red color letters help distinguish between the source and target MDPs.

(+) The implementation of the method is also discussed, making the method applicable. The hyperparameter settings are reported in the appendix.

In the meanwhile, there exist concerns and limitations:

(-) My concern comes from the experiment section. The experiment results are averaged over 3 seeds only, which does not give an accurate estimation of the true performance. Even though there are 4 environments with 4 dynamic shift cases each and DARA improves the performance in most cases, the risk still exists that the averaged performance of one single test is inaccurate. Therefore it could be better if more random seeds can be tested.

(-) It seems like the proposed method is limited to the case that the target environment is restricting transitions that exist in the source task. It is not clear to me if the method still helps when the target MDP adds new transitions. For example, in the Map task, collecting source dataset with the obstacle and collecting target dataset without the obstacle. In this case, it can be harder for the framework to capture the change in the transition dynamics, thus harder for the agent to figure out the new optimal policy (the shorter path without the obstacle). If this is true, the cases that DARA can be applied to might be limited, though I believe it is still helpful in cases like proposed in the paper.


**Summary Of The Paper:**

The paper focuses on the dynamics shift problem in the offline reinforcement learning setting, with the assumption that the estimated MDP from the offline dataset is deterministic. It proposes DARA, a framework that relaxes the requirement of data (both the amount of data and the reward information contained in the data) needed from the target task. By capturing the dynamics shift between the source and target environments, DARA modifies the reward gained in the source environment to discourage transitions that have a smaller probability to happen in the target environment. The framework works with both model-free and model-based settings.

**Summary Of The Review:**

I recommend acceptance. The paper introduces a framework to adapt to the target environment with a relaxed requirement on data from the target task. I believe this work contributes to improving the data efficiency in the offline reinforcement learning setting, even when the access of the reward function in the target environment is limited. There exists shortages and limitations as listed, but overall, I consider the positives to outweigh the negatives.

---

> ### Author Response · Authors · 2021-11-15
> **Author Response**
>
> We would like to thank the reviewer for your time and thoughtful comments! Please see our response below.
>
> **(1) "It could be better if more random seeds can be tested."**
>
> Thank you for your suggestion. We have added more random seeds (from 3 seeds to 5 seeds) over the tasks and provided corresponding error bars in the appendix (Tables 21 and 22).  The new results also support our conclusion in the main text regarding the impact of reduced target offline data and the effectiveness of our dynamics-aware reward modification.
>
> **(2) "It seems like the proposed method is limited to the case that the target environment is restricting transitions that exist in the source task. It is not clear to me if the method still helps when the target MDP adds new transitions. ..., thus harder for the agent to figure out the new optimal policy (the shorter path without the obstacle)."**
>
> Thank you for the insightful comment. If we directly perform DARA *with only the source offline data* $\textcolor{blue}{\mathcal{D'}}$, this limitation does exist, where the learned behaviors will be restricted to the source offline data. However, as stated in Algorithm 1, we perform offline RL algorithms *with both target offline data and source offline data* $\\{ \textcolor{red}{\mathcal{D}} \ \cup \ \textcolor{blue}{\mathcal{D'}} \\}$. Thus, to some extent, such limitation can be overcome as long as offline RL algorithm captures the information (eg. the short path without the obstacle) contained in the (limited) target offline data $\textcolor{red}{\mathcal{D}}$.
>
> In Figure 13 in appendix A.6 (in our revised paper), we show the illustration regarding whether the learned policy is limited to the source offline data, where we exchange the source environment and the target environment in Figure 2 (in the main text). We can find that if we directly perform DARA with only the source offline data $\textcolor{blue}{\mathcal{D'}}$ (ie. 0k in Figure 13), we indeed can not acquire the optimal trajectory (eg. the short path without the obstacle). However, as we increase the number of target offline data $\textcolor{red}{\mathcal{D}}$, training with $\\{ \textcolor{red}{\mathcal{D}} \ \cup \ \textcolor{blue}{\mathcal{D'}} \\}$ can gradually acquire optimal trajectories.
>
> We hope the above response addresses the reviewer’s concerns and welcome any further feedback.

---

> > ### Comment · Reviewer_QyWv · 2021-11-16
> > **Thank you for your response**
> >
> > I would like to thank the authors for adding more results and answering my questions. The response addressed my concerns. I would like to maintain the recommendation for acceptance.

---

> > > ### Author Response · Authors · 2021-11-17
> > > **Thank you for the response!**
> > >
> > > We really appreciate your comments!

---

### Official Review · Reviewer_wSxQ · 2021-11-02

**Correctness:** 3
**Technical Novelty And Significance:** 3
**Empirical Novelty And Significance:** 4
**Recommendation:** 8
**Confidence:** 4

**Main Review:**

## Pros
* The proposed framework is well justified for both model-free and model-based approaches. And the framework is easy to implement.
* Figure 3 clearly shows that the reward penalty obtained through domain classifiers can aid the knowledge transfer when the dynamics shift.
* The propose method has been tested on real environment and show improvement over the baseline aggregation strategy. This highlights the importance of dynamics-aware transfer in offline transfer RL for practical applications.

## Cons/questions/suggestions
* __Relationship to importance sampling__ The proposed reward compensation term is strongly related to importance weights. What if one simply weight each of the original offline data point based on the importance weights and train a policy using the combined importance weights source dataset and target dataset? Do you expect any advantage of the proposed method over importance weighting?
* __Explanation for Lemma 1__ Could the authors elaborate on Lemma 1? Even if the environment has deterministic transition dynamics, given a finite randomly sampled dataset $\mathcal{D}$, the learned dynamics $\hat{T}$ model could still be different from the true dynamics $T$. Consider a simple tabular setting where there is a single initial state, a single allowed action, and two different states that could transition from this initial state and the only allowed action with equal probability. If I was given a dataset with ten state-action-next state tuples, and four of them correspond to the transition state 1 while the other six contain the transition state 2. Then I will get a wrong estimate of the environment dynamics. Please correct me if my understanding of the dynamic shift is incorrect.
* __Impact of offline dataset quality__ Given different offline dataset quality, the proposed framework's improvement over the simple aggregation of source and target should vary. It would be nice for authors to compute the average improvement for all six algorithms over Hoper and Walker2d and see if there's any consistent trend. If so, it is important to discuss what insights we could learn from the results.

## Minor comments
* Could the authors comment on the sensitivity of the algorithm on the hyperparameter $\eta$ in $r_{t} \leftarrow r_{t}-\eta \Delta r$? How would one pick the value of $\eta$ in practice?

**Summary Of The Paper:**

Offline reinforcement learning could enable one to train a data-driven decision making engine from purely offline data. However, the performance of polices learned through existing offline RL methods heavily depend on the quantity and quality of the offline data. When the data is scarce for a task, existing offline RL methods trained on limited offline data could perform poorly.

This work tackles the data efficiency issue of offline RL by transferring knowledge from a source offline dataset to a target offline dataset following an environment dynamics different from the source offline dataset. The proposed method augments the reward function by penalizing the state-action-next state transition tuples that are rare in the source environment. This reward augmentation strategy directly accounts for the dynamics shift between the source and the target domain. On several simulation a real-world tasks, the proposed method significantly improved the data-efficiency of existing offline RL algorithms.

**Summary Of The Review:**

This well-written paper studies the important problem of transferring knowledge to improve sample efficiency for offline RL. The proposed reward augmentation framework is well justified for both the model-free and model-based settings from the perspective of variational lower bound and model-bias when the dynamics shift, respectively. Also, the experiment results support authors' claims well. Overall, this is a good submission that can inspire future works along this direction.

---

> ### Author Response · Authors · 2021-11-15
> **Author Response**
>
> Thank you for the great summarization of our main contributions and we really appreciate your encouraging comments. Please see our response below.
>
> **(1) Relationship to importance sampling.**
>
> Thank the reviewer for raising importance sampling (IS), which is a simple strategy for incorporating off-policy (or off-dynamics) data. However, this approach (IS) is prone to high variance and instability [1,2,3], which, in practice, requires a variety of stabilization techniques to ensure consistent performance. Following [3], we thus use the logarithm of the density ratio to modify the reward. We also report the experimental comparison between DARA and IS-based dynamics adaption in Table 24 in our revised paper. We can find that in most of the tasks, our DARA performs better than the IS-based approaches.
>
> [1] Peng, Xue Bin, et al. "Advantage-weighted regression: Simple and scalable off-policy reinforcement learning." (2019). \\\
> [2] Nachum, Ofir, et al. "Algaedice: Policy gradient from arbitrary experience." (2019). \\\
> [3] Eysenbach, Benjamin, et al. "Off-Dynamics Reinforcement Learning: Training for Transfer with Domain Classifiers." (2021).
>
>
>
>
> **(2) Explanation for Lemma 1.**
>
>
> The example scenario described by the reviewer does illustrate the problem of dynamics shit. However, one assumption in Lemma 1 is that the transition dynamics are deterministic, that is, there is a real-valued function $T(s, a)$ such that when state $s$ and action $a$ are chosen, then the next state is fixed: $s' = T(s, a)$. However, the example described by the reviewer is under *stochastic transition dynamics*. We think that in a stochastic environment, there is always a dynamics shift. Thus, we assume that the environment is deterministic, which is also consistent with [4].
>
> [4] Fujimoto, Scott, David Meger, and Doina Precup. "Off-policy deep reinforcement learning without exploration." (2019).
>
> **(3) Impact of offline dataset quality.**
>
> That is an excellent point! We provide the average improvement (over 1T+10S w/o Aug.) for all algorithms over Hopper and Walker2d, as shown in the following table.
>
> |               |     Hopper      |      Hopper       |    Walker2d     |     Walker2d      |
> | ------------- | :-------------: | :---------------: | :-------------: | :---------------: |
> |               | Body Mass Shift | Joint Noise Shift | Body Mass Shift | Joint Noise Shift |
> | Random        |      25.6%      |       23.3%       |      30.4%      |      115.0%       |
> | Medium        |      54.1%      |       23.6%       |      10.0%      |       5.6%        |
> | Medium-Replay |      62.4%      |       74.4%       |     100.6%      |       62.1%       |
> | Medium-Expert |      37.5%      |       35.4%       |      56.5%      |       19.2%       |
>
> We can find that the performance improvement on the Medium-Replay is the highest (except for Walker2d with joint noise shift). We speculate that this is due to two reasons: 1) The performance of adopted offline RL baselines basically depends on the quality of offline data (the more expert data, the better performance); 2) The reward modification requires offline data to have a certain exploration ability in the environment, otherwise the data lacking exploration ability will lead to too conservative dynamics correction. These two requirements can be guaranteed in Medium-Replay data: "medium" data provides optimal/suboptimal behaviors; "replay" data provides exploration.
>
> **(4) Hyper-parameter $\eta$.**
>
> Please refer to Question (2) of Reviewer 73ni.
>
> We hope the above response addresses the reviewer’s concerns and welcome any further feedback.

---

> > ### Comment · Reviewer_wSxQ · 2021-11-15
> > **Thanks for the comments.**
> >
> > I'd like to thank the authors for answering my questions and adding the additional results on importance sampling and dataset quality. Since these additional results are not major changes, I will maintain my original rating for this submission.

---

> > > ### Author Response · Authors · 2021-11-15
> > > **Thank you for the response!**
> > >
> > > We really appreciate your comments!

---

### Official Review · Reviewer_73ni · 2021-11-04

**Correctness:** 3
**Technical Novelty And Significance:** 2
**Empirical Novelty And Significance:** 2
**Recommendation:** 5
**Confidence:** 5

**Main Review:**

Pros:
1. The paper is clearly written and easy to understand.
2. DARA appears to be the first method that addresses the problem of domain adaptation in the offline RL setting.
3. The authors show extensive empirical results of DARA on both simulated and real-world experiments and evaluate the performance of DARA combined with most of the prior offline RL methods and observe reasonable improvement.

Cons:
1. I think the algorithm is a bit incremental. It is in some sense a combination of Eysenbach et al. 2021 and conservative penalties introduced in prior model-based and model-free works. It seems to be a bit straightforward and therefore the novelty is a bit limited.
2. DARA introduced additional hyperparameters compared to prior offline RL papers, i.e. the coefficient that controls dynamics shift, which could make the method hard to tune. DARA also requires training two classifiers, which also add another layer of complexity in terms of reproducibility. I also didn't see the mention of the hyperparameter settings. It would be important to know how the authors select hyperparameters for all of the environments since hyperparameter tuning in offline RL is usually expensive and potentially dangerous. It is also important to see the error bars of the results.
3. The real-world experiments seem to use state inputs. It would be really intriguing if the authors could show good transfer results with image inputs in the sim-to-real setting.

**Summary Of The Paper:**

This paper presents a new offline RL algorithm that aims to adapt the policy to a target domain with insufficient data with the help of abundant data in a source domain. The authors propose their method, DARA, which applies a reward penalty that correct the dynamics shift between the source and target domains. The authors combine DARA with previous model-based and model-free offline RL methods and show that DARA is able to achieve performance improvement over prior offline RL methods without the reward augmentation in both simulated and real-world tasks.

**Summary Of The Review:**

I think the paper proposes a new algorithm that tackles a somewhat new problem, but the methodology lacks originality and the hyperparameter details are missing. Therefore, I would vote for a weak reject.

---

> ### Author Response · Authors · 2021-11-15
> **Author Response**
>
> Thank you for the comments and suggestions! Please see our response below.
>
> **(1) "I think the algorithm is a bit incremental."**
>
> Our work is motivated by [1,2,3], which inspires us to use reward modification to correct the dynamics shift. Compared to [3] (Eysenbach et al. 2021) mentioned by the reviewer, there are a few key differences:
>
> (1) Theory. We introduce a unified formalism for the dynamics shift in both model-free and model-based (offline) settings, while Eysenbach et al. 2021 only focuses on the model-free (online) setting.
>
> (2) Algorithm. We propose a reward modification to the offline training with explicit policy constraints as well as implicit policy constraints, while Eysenbach et al. 2021 performs training only with explicit policy constraints.
>
> (3) Experiment. We demonstrate our method in both simulated and real-world tasks, while all the tasks in Eysenbach et al. 2021 are simulated.
>
> (4) RL Community. For online RL, there has been plenty of work on domain (dynamics) adaptation. For offline RL, to date, domain (dynamics) adaptation receives little attention. We also provide lots of experimental details (especially for quadruped robot tasks) and offline data (with dynamics shift), as a complement to D4RL dataset.
>
> |                               | [1]        | [2]        | [3]        | DARA                     |
> | ----------------------------- | ---------- | ---------- | ---------- | ------------------------ |
> | Online/Offline RL             | Online     | Online     | Online     | Offline                  |
> | Model-free/Model-based        | Model-free | Model-free | Model-free | Model-free + Model-based |
> | Explicit/Implicit Constraints | Explicit   | Explicit   | Explicit   | Explicit + Implicit      |
> | Experiment                    | Simulated  | Simulated  | Simulated  | Simulated + Real-world   |
> | Community (Related Work)      | Plenty     | Plenty     | Plenty     | Little                   |
>
> [1] Eysenbach, Benjamin, and Sergey Levine. "Maximum entropy rl (provably) solves some robust rl problems." (2021). \\\
> [2] Kumar, Saurabh, et al. "One Solution is Not All You Need: Few-Shot Extrapolation via Structured MaxEnt RL." (2020). \\\
> [3] Eysenbach, Benjamin, et al. "Off-Dynamics Reinforcement Learning: Training for Transfer with Domain Classifiers." (2021).
>
> **(2) Hyper-parameters (the coefficient that controls dynamics shift) and error bars.**
>
> Hyper-parameters $\eta$ : We report that we set $\eta = 0.1$ for all simulated tasks and set $\eta = 0.01$ for the sim2real task. We also examine the sensitivity of DARA on the hyper-parameters $\eta$. We refer the reviewer to Table 23 (Appendix A.6) in our revised paper.
>
> Error bars: We continue to add two random seeds (from 3 seeds to 5 seeds), and provide corresponding error bars in the appendix (Tables 21 and 22).  The new results also support our conclusion in the main text regarding the impact of reduced target offline data and the effectiveness of our reward modification. We refer the reviewer to our revised paper for the new results.
>
> **(3) The real-world experiments seem to use state inputs. It would be really intriguing if the authors could show good transfer results with image inputs in the sim-to-real setting.**
>
> Thank the reviewer for raising the sim2real task with image inputs, but we respectfully disagree that this experiment is necessary for this paper.
>
> First, for learning quadrupedal locomotion, the state inputs are more realistic than the high-dimensional image inputs (because the joint (state) information of quadruped robot is often accessible). Generally speaking, visual information is considered when a robot is faced with complex high-level tasks. We aim to learn locomotion skills, not complex high-level decisions, so only state information is considered. Such consideration is also consistent with much of the current work around quadruped robots, such as the notable [4, 5].
>
> Second, low-fidelity simulated sensors like image renderers (in simulator) are often unable to reproduce the richness and noise produced by their real-world counterparts [6]. Such sim2real transfer will thus characterize both the dynamics shift and the state representation shift between the simulator and the real-world, which is exceptionally difficult. We respectfully argue that this transfer task (with image inputs) is out of scope for this work. Considering that such state representation shift is orthogonal to our dynamics shift, we leave this as future work.
>
> [4] Lee, Joonho, et al. "Learning quadrupedal locomotion over challenging terrain." (2020). \\\
> [5] Kumar, Ashish, et al. "Rma: Rapid motor adaptation for legged robots." (2021). \\\
> [6] Tobin, Josh, et al. "Domain randomization for transferring deep neural networks from simulation to the real world." (2017).
>
> We hope the above response addresses the reviewer’s concerns and welcome any further feedback.

---

> > ### Comment · Reviewer_73ni · 2021-12-03
> > **Thank you for the reply; Additional comment**
> >
> > Thank you for the response! It addressed most of my concerns. However, I'm still unsure if the method is novel enough as it appears to be simply Eysenbach et al. 2021 combined with standard offline RL methods. It is true that you added model-based analysis as well as implicit policy constraints, but these new components seems to be directly adapted from existing works. It is also unclear whether explicit or implicit constraints are used in the experiments. Therefore, I still have concern on the novelty of the paper.

---

> > > ### Author Response · Authors · 2021-12-03
> > > **Author Response**
> > >
> > > Thank you for the reply! Here, we will address your remaining concerns.
> > >
> > > **(1) Compared with Eysenbach et al. 2021**
> > >
> > > In addition to the differences in problem setup, derivation, experiment, and research background (as mentioned in the above response), we provide two detailed differences from *updating the policy* and *learning reward modification*.
> > >
> > > **(Updating the policy)** One key difference is that we use *both* the source and target data to update the policy, while Eysenbach et al. 2021 *only* adopts the data in source domain to update the policy. *This difference is non-trivial.* If we combine Eysenbach et al. 2021 and offline RL methods strictly, updating policy only with the source offline data, this will lead to the issue mentioned by Reviewer QyWv --- learning will be limited to the data in the source domain. However, we use both source and target domain data to update the policy, which can alleviate the issue mentioned by Reviewer QyWv (see Figure 13 in appendix A.6).
> > >
> > > **(Learning reward modification)** Compared with Eysenbach et al. 2021, one main similarity between us is that we adopt the same classifiers to correct for the dynamics shift. The main reason we adopt the classifiers is for the generality of the framework. In Appendix A.3.2, we also consider that we do not learn the classifiers and choose to directly learn the transition probability to correct for the dynamics shift. The difference in performance between the two (classifiers vs. learned transition ratio) is not significant, *so experimentally it is desirable to learn the transfer probability (ratio) directly*. We use the classifiers for the generality of the method (especially for model-free formulation or given visual offline data). We respectfully argue that it would be inappropriate to associate Eysenbach et al. 2021 too much because we use the same classifiers.
> > >
> > > **(2) It is also unclear whether explicit or implicit constraints are used in the experiments.**
> > >
> > > In the experiment, we use implicit constraints for both MPO and AWR, and explicit constraints for other baselines. Sorry for not specifying this, we will add the corresponding description.
> > >
> > > Again, thank you for helping to improve the paper! We hope the above response addresses the reviewer’s concerns, but please let us know if you have any additional questions or concerns.

---

### Decision · Program_Chairs · 2022-01-20

**Decision:**

Accept (Poster)

**Comment:**

**Summary**

This paper proposes a method to do a sample-efficient offline domain adaptation method where the setting requires one to have abundant amount of data from the source domain but limited amount of data available in the target domain. The proposed approach DARA achieves that by accounting for the the dynamics shift between the source and the target domain via a reward penalty. The paper shows promising experimental results.

**Final Thoughts**
Overall the paper is well-written, and it addresses an important problem. The reviewers are mostly positive about the paper at the end. The authors did a very good job addressing the concerns raised by the reviewers. Reviewer 73ni had concerns about the novelty of the approach, it would be nice if the paper can make it more clear about the novelty of the proposed approach compared to the other existing methods in the paper. I  would also recommend the authors to incorporate the feedback and the suggestions made by the reviewer into the camera-ready version of the paper.